# Realizable $\mathcal{H}$-Consistent and Bayes-Consistent Loss Functions for Learning to Defer

**Anqi Mao**
Courant Institute
New York, NY 10012
aqmao@cims.nyu.edu

**Mehryar Mohri**
Google Research & CIMS
New York, NY 10011
mohri@google.com

**Yutao Zhong**
Courant Institute
New York, NY 10012
yutao@cims.nyu.edu

## Abstract

We present a comprehensive study of surrogate loss functions for learning to defer. We introduce a broad family of surrogate losses, parameterized by a non-increasing function $\Psi$, and establish their realizable $\mathcal{H}$-consistency under mild conditions. For cost functions based on classification error, we further show that these loss functions admit $\mathcal{H}$-consistency bounds when the hypothesis set is symmetric and complete, a property satisfied by common neural network and linear function hypothesis sets. Our results also resolve an open question raised in previous work [Mozannar et al., 2023] by proving the realizable $\mathcal{H}$-consistency and Bayes-consistency of a specific surrogate loss. Furthermore, we identify choices of $\Psi$ that lead to $\mathcal{H}$-consistent surrogate losses for *any general cost function*, thus achieving Bayes-consistency, realizable $\mathcal{H}$-consistency, and $\mathcal{H}$-consistency bounds *simultaneously*. We also investigate the relationship between $\mathcal{H}$-consistency bounds and realizable $\mathcal{H}$-consistency in learning to defer, highlighting key differences from standard classification. Finally, we empirically evaluate our proposed surrogate losses and compare them with existing baselines.

## 1 Introduction

In many practical scenarios, combining expert insights with established models can yield significant enhancements. These experts can be human domain specialists or more complex, albeit resource-intensive, models. For example, modern language and dialogue models are prone to producing *hallucinations* or inaccurate information. The quality of their responses can be significantly enhanced by delegating uncertain predictions to more specialized or advanced pre-trained models. This problem is particularly crucial for large language models (LLMs), as noted in [Wei et al., 2022, Bubeck et al., 2023]. The same principle applies to other generative systems, like those for images or videos, and to learning models in diverse applications such as image classification, annotation, and speech recognition. Thus, the task of *learning to defer* (L2D) with experts has become increasingly critical across a wide array of applications.

Directly optimizing the deferral loss function, which is the target loss in L2D, is computationally intractable for many choices of the hypothesis set. Therefore, a common approach is to optimize a surrogate loss that facilitates the optimization of the deferral loss function. Recent work in L2D has proposed several surrogate losses [Mozannar and Sontag, 2020, Verma and Nalisnick, 2022, Mozannar et al., 2023, Mao et al., 2024a] and studied their consistency guarantees, including Bayes-consistency, realizable $\mathcal{H}$-consistency, and $\mathcal{H}$-consistency bounds (see definitions in Section 3.2). In particular, Mozannar and Sontag [2020] proposed the first Bayes-consistent surrogate loss by generalizing the cross-entropy loss for L2D. Verma and Nalisnick [2022] proposed an alternative Bayes-consistent surrogate loss by generalizing the one-versus-all loss for L2D. Mozannar et al. [2023] showed that these surrogate losses are not realizable $\mathcal{H}$-consistent. They proposed an alternative surrogate loss

that is realizable $\mathcal{H}$-consistent, but they were unable to prove or disprove whether the proposed surrogate loss is Bayes-consistent. All the surrogate losses mentioned above and their consistency guarantees hold only for cost functions based on classification error. Mao et al. [2024a] generalized the surrogate loss in [Mozannar and Sontag, 2020] to incorporate general cost functions and any multi-class surrogate losses. They provided $\mathcal{H}$-consistency bounds for the novel family of surrogate losses, offering a stronger guarantee than Bayes-consistency.

However, none of these surrogate losses satisfies all these guarantees simultaneously. In particular, a recent AISTATS notable award paper by Mozannar et al. [2023] left open the problem of finding surrogate losses that are both Bayes-consistent and realizable $\mathcal{H}$-consistent when the cost function for the expert is its classification error. The problem becomes even more challenging when considering more general and realistic cost functions.

We present a comprehensive analysis of surrogate loss functions for L2D. Our contributions address the limitations of previous approaches and provide a unified framework for designing surrogate losses with strong theoretical guarantees. In Section 4, we first introduce a broad family of surrogate losses for L2D, derived from first principles (Section 4.1). This family is parameterized by a non-increasing function $\Psi$, which provides some flexibility in tailoring the loss function to specific requirements. We establish that under mild conditions on $\Psi$, these surrogate losses achieve realizable $\mathcal{H}$-consistency, a key guarantee for many applications (Section 4.2).

Next, for cost functions based on classification error, we further establish that our surrogate loss functions admit $\mathcal{H}$-consistency bounds when the hypothesis set is symmetric and complete (Section 4.3). This result holds for commonly used neural network and linear function hypothesis sets, further strengthening the applicability of our results. Additionally, our results resolve an open question raised by Mozannar et al. [2023] by proving the realizable $\mathcal{H}$-consistency and Bayes-consistency of their proposed surrogate loss, which the authors had left as an open question (Section 4.4).

In Section 4.3, we further identify specific choices of $\Psi$, such as the one corresponding to the mean absolute error loss, that lead to $\mathcal{H}$-consistent surrogate losses for *any general cost function*. These loss functions are adapted to general cost functions and benefit from Bayes-consistency (Section 4.4), realizable $\mathcal{H}$-consistency, and $\mathcal{H}$-consistency bounds *simultaneously*.

In Section 5, we also study the relationship between $\mathcal{H}$-consistency bounds and realizable $\mathcal{H}$-consistency in the context of L2D, highlighting key distinctions from the standard classification setting. Finally, we further report the results of experiments with our new surrogate losses and their comparison with the baselines in different settings (Section 6).

We discuss the related work in Section 2 and then begin with the preliminaries in Section 3.

## 2 Related work

The approach of *single-stage learning to defer*, where a predictor and a deferral function are trained together, was pioneered by Cortes, DeSalvo, and Mohri [2016a,b, 2023] and further developed in subsequent studies on abstention, where the cost is constant [Charoenphakdee et al., 2021, Cao et al., 2022, Li et al., 2023, Cheng et al., 2023, Mao et al., 2024c,b, Mohri et al., 2024] and on deferral, where the cost can vary depending on the instance and the label [Mozannar and Sontag, 2020, Verma and Nalisnick, 2022, Mozannar et al., 2023, Verma et al., 2023, Cao et al., 2023, Mao et al., 2023a, 2024a]. In this approach, the deferral function determines whether to defer to an expert for each input. This approach has been shown to be superior to *confidence-based* approaches, where the decision to abstain or defer is based solely on the magnitude of the predictor's value [Chow, 1957, 1970, Bartlett and Wegkamp, 2008, Yuan and Wegkamp, 2010, 2011, Ramaswamy et al., 2018, Ni et al., 2019, Jitkrittum et al., 2023]; and to *selective classification* approaches, where the selection rate is fixed and a cost function modeled by an expert cannot be taken into account [El-Yaniv et al., 2010, El-Yaniv and Wiener, 2012, Wiener and El-Yaniv, 2011, 2012, 2015, Geifman and El-Yaniv, 2017, 2019, Acar et al., 2020, Gangrade et al., 2021, Zaoui et al., 2020, Jiang et al., 2020, Shah et al., 2022].

Madras et al. [2018] initiated the *learning to defer* (L2D) problem scenario, which integrates human expert decisions into the cost function. This approach has been further explored in subsequent studies [Raghu et al., 2019, Wilder et al., 2021, Pradier et al., 2021]. Mozannar and Sontag [2020] introduced the first Bayes-consistent surrogate loss for L2D, which was further refined in [Raman and Yee, 2021, Liu et al., 2022]. Verma and Nalisnick [2022] proposed an alternative Bayes-consistent surrogate loss,

the one-versus-all loss, which was later examined within a broader family of loss functions [Charusaie et al., 2022]. Cao et al. [2023] proposed an asymmetric softmax function, which can induce a valid probability estimator for learning to defer. Mozannar et al. [2023] showed that the surrogate losses in [Mozannar and Sontag, 2020, Verma and Nalisnick, 2022] are not realizable $\mathcal{H}$-consistent. They proposed an alternative surrogate loss that is realizable $\mathcal{H}$-consistent, but they were unable to prove or disprove whether the proposed surrogate loss is Bayes-consistent. All the surrogate losses mentioned above and their consistency guarantees hold only for cost functions based on classification error. Mao et al. [2024a] generalized the surrogate loss in [Mozannar and Sontag, 2020] to incorporate general cost functions and any multi-class surrogate losses. They provided $\mathcal{H}$-consistency bounds for the novel family of surrogate losses, offering a stronger guarantee than Bayes-consistency.

Additional studies have focused on post-hoc methods, with Okati et al. [2021] suggesting an alternative optimization technique between the predictor and rejector, and Narasimhan et al. [2022] offering corrections for underfitting surrogate losses [Liu et al., 2024], and Charusaie and Samadi [2024] providing a unifying post-processing framework for multi-objective L2D based on a generalization of the Neyman-Pearson Lemma [Neyman and Pearson, 1933]. The L2D framework or variations thereof have found applications in diverse scenarios, spanning regression, reinforcement learning, and human-in-the-loop systems, among others [De et al., 2020, 2021, Straitouri et al., 2021, Zhao et al., 2021, Joshi et al., 2021, Gao et al., 2021, Mozannar et al., 2022, Hemmer et al., 2023, Chen et al., 2024, Palomba et al., 2024]. More recently, the problem of *learning to defer with multiple experts* has been analyzed in several publications [Hemmer et al., 2022, Keswani et al., 2021, Kerrigan et al., 2021, Straitouri et al., 2022, Benz and Rodriguez, 2022, Verma et al., 2023, Mao et al., 2023a, 2024a,g, Tailor et al., 2024]. Meanwhile, Mao et al. [2023a] also proposed a *two-stage learning to defer* framework. They introduced two-stage surrogate losses that are both Bayes-consistent and realizable $\mathcal{H}$-consistent with constant costs. However, realizable $\mathcal{H}$-consistency does not hold for cost functions based on classification error. As with [Mozannar and Sontag, 2020, Verma and Nalisnick, 2022, Mozannar et al., 2023], our work focuses on the single-stage and single-expert setting, and we plan to explore a similar approach in a multi-expert/two-stage setting in the future.

## 3 Preliminaries

We start with the definitions and notations used in the learning-to-defer scenario considered in this paper. We will then introduce consistency guarantees, including *Bayes consistency*, *Realizable $\mathcal{H}$-consistency*, and $\mathcal{H}$-*consistency bounds*. Finally, we will review existing consistent surrogate losses for L2D.

### 3.1 Learning to defer: problem setup

Let $\mathcal{X}$ be an input space and $\mathcal{Y} = [n] \coloneqq \{1, \ldots, n\}$ be the label space in the standard multi-class classification setting. We study the *learning to defer* (L2D) scenario, where a learner can either predict a label from $\mathcal{Y}$ or defer to an expert.

To model this, we introduce an augmented label space $\overline{\mathcal{Y}} = \{1, \ldots, n, n + 1\}$, where the label $n + 1$ corresponds to deferral. An expert is a fixed predictor $g \colon \mathcal{X} \times \mathcal{Y} \to \mathbb{R}$. The goal of L2D is to select a predictor $h$ out of a hypothesis set $\mathcal{H}$ of functions mapping from $\mathcal{X} \times \overline{\mathcal{Y}}$ to $\mathbb{R}$ with small expected *deferral loss*. Let $\mathsf{h}(x)$ denote the prediction of $h$ on input $x \in \mathcal{X}$, defined as $\mathsf{h}(x) = \operatorname{argmax}_{y \in \overline{\mathcal{Y}}} h(x, y)$, that is the label in the augmented label space $\overline{\mathcal{Y}}$ with the highest score, with an arbitrary but fixed deterministic strategy for breaking ties. Then, the *deferral loss function* $\mathsf{L}_{\mathrm{def}}$ is defined as follows:

$$\forall (x, y) \in \mathcal{X} \times \mathcal{Y}, \quad \mathsf{L}_{\mathrm{def}}(h, x, y) = 1_{\mathsf{h}(x) \neq y} 1_{\mathsf{h}(x) \in [n]} + c(x, y) 1_{\mathsf{h}(x) = n+1},$$

where $c(x, y)$ is the the cost of deferring on input $x$ with true label $y$. If the deferral option is selected, that is $\mathsf{h}(x) = n + 1$, the deferral cost $c(x, y)$ is incurred. Otherwise, the prediction of $h$ is within the standard label space, $\mathsf{h}(x) \in [n]$, and the loss incurred coincides with the standard zero-one classification loss, $1_{\mathsf{h}(x) \neq y}$.

The choice of the cost function $c$ is flexible. For example, the cost can be defined as the expert's classification error: $c(x, y) = 1_{\mathsf{g}(x) \neq y}$, as in previous work [Mozannar and Sontag, 2020, Verma and Nalisnick, 2022, Mozannar et al., 2023]. Here, $\mathsf{g}(x) = \operatorname{argmax}_{y \in [n]} g(x, y)$ is the prediction made by

the expert $g$. More generally, it can incorporate the inference cost for the expert [Mao et al., 2024a]: $c(x, y) = \alpha 1_{g(x) \neq y} + \beta$, with $\alpha, \beta > 0$. We assume, without loss of generality, that the cost is bounded by 1: $0 \leq c(x, y) \leq 1$, which can be achieved through normalization in practice.

## 3.2 Consistency guarantees

Directly optimizing the deferral loss function, which is the target loss in L2D, is generally computationally intractable for complex hypothesis sets $\mathcal{H}$. Therefore, a common approach is to optimize a surrogate loss that facilitates the optimization of the deferral loss function. A natural learning guarantee for such surrogate losses is *Bayes-consistency* [Zhang, 2004a, Bartlett et al., 2006, Zhang, 2004b, Tewari and Bartlett, 2007, Steinwart, 2007]:

**Definition 3.1** (Bayes-consistency). A surrogate loss $\mathsf{L}$ is Bayes-consistent with respect to $\mathsf{L}_{\mathrm{def}}$, if minimizing the surrogate loss over the family of all measurable functions leads to the minimization of the deferral loss:

$$\lim_{n \to +\infty} \mathcal{E}_{\mathsf{L}}(h_n) - \mathcal{E}_{\mathsf{L}}^*(\mathcal{H}_{\mathrm{all}}) = 0 \implies \lim_{n \to +\infty} \mathcal{E}_{\mathsf{L}_{\mathrm{def}}}(h_n) - \mathcal{E}_{\mathsf{L}_{\mathrm{def}}}^*(\mathcal{H}_{\mathrm{all}}) = 0.$$

Here, given a distribution $\mathcal{D}$ over $\mathcal{X} \times \mathcal{Y}$ and a loss function $\mathsf{L} \colon \mathcal{H} \times \mathcal{X} \times \mathcal{Y} \to \mathbb{R}$, we denote by $\mathcal{E}_{\mathsf{L}}(h)$ the *generalization error* of a hypothesis $h \in \mathcal{H}$, $\mathcal{E}_{\mathsf{L}}(h) = \mathbb{E}_{(x,y) \sim \mathcal{D}}[\mathsf{L}(h, x, y)]$, and by $\mathcal{E}_{\mathsf{L}}^*(\mathcal{H})$ the *best-in-class generalization error*, $\mathcal{E}_{\mathsf{L}}^*(\mathcal{H}) = \inf_{h \in \mathcal{H}} \mathcal{E}_{\mathsf{L}}(h)$. Bayes-consistency assumes that the optimization occurs over the family of all measurable functions, $\mathcal{H}_{\mathrm{all}}$. However, in practice, the hypothesis set of interest is typically a restricted one, such as a family of neural networks. Therefore, a hypothesis-dependent learning guarantee, such as $\mathcal{H}$-*consistency bounds* [Awasthi et al., 2022a,b] (see also [Awasthi et al., 2021a,b, 2023, 2024, Mao et al., 2023b,e,f, Zheng et al., 2023, Mao et al., 2023c,d, 2024h,e,d,f, Cortes et al., 2024]) and *realizable $\mathcal{H}$-consistency* [Long and Servedio, 2013, Zhang and Agarwal, 2020], is more informative and relevant. Realizable $\mathcal{H}$-consistency, defined as follows, requires that a minimizer of the surrogate loss over the given hypothesis set $\mathcal{H}$ also minimizes the target loss, provided that the underlying distribution is realizable.

**Definition 3.2** (Realizable $\mathcal{H}$-consistency). A surrogate loss $\mathsf{L}$ is realizable $\mathcal{H}$-consistent with respect to $\mathsf{L}_{\mathrm{def}}$, if for any distribution over which there exists a predictor $h^* \in \mathcal{H}$ achieving zero deferral loss, $\mathcal{E}_{\mathsf{L}_{\mathrm{def}}}(h^*) = 0$, minimizing the surrogate loss also leads to a zero-error solution:

$$\hat{h} \in \operatorname*{argmin}_{h \in \mathcal{H}} \mathcal{E}_{\mathsf{L}}(h) \implies \mathcal{E}_{\mathsf{L}_{\mathrm{def}}}(\hat{h}) = 0.$$

Note that realizable $\mathcal{H}$-consistency does not imply Bayes-consistency, even if we set $\mathcal{H} = \mathcal{H}_{\mathrm{all}}$ in Definition 3.2, since Bayes-consistency requires that the relationship holds for all distributions, not just realizable ones. $\mathcal{H}$-*consistency bounds*, on the other hand, always imply Bayes-consistency. Given a hypothesis set $\mathcal{H}$, a surrogate loss $\mathsf{L}$ admits an $\mathcal{H}$-*consistency bound*, if for some non-decreasing concave function $\Gamma \colon \mathbb{R}_+ \to \mathbb{R}_+$ with $\Gamma(0) = 0$, a bound of the following form holds for any hypothesis $h \in \mathcal{H}$ and any distribution:

$$\mathcal{E}_{\mathsf{L}_{\mathrm{def}}}(h) - \mathcal{E}_{\mathsf{L}_{\mathrm{def}}}^*(\mathcal{H}) + \mathcal{M}_{\mathsf{L}_{\mathrm{def}}}(\mathcal{H}) \leq \Gamma(\mathcal{E}_{\mathsf{L}}(h) - \mathcal{E}_{\mathsf{L}}^*(\mathcal{H}) + \mathcal{M}_{\mathsf{L}}(\mathcal{H})), \tag{1}$$

where $\mathcal{M}_{\mathsf{L}}(\mathcal{H})$ is *the minimizability gap*, defined as the difference between the best-in-class generalization error and the expected pointwise infimum loss: $\mathcal{M}_{\mathsf{L}}(\mathcal{H}) = \mathcal{E}_{\mathsf{L}}^*(\mathcal{H}) - \mathbb{E}_x[\inf_{h \in \mathcal{H}} \mathbb{E}_{y|x}[\mathsf{L}(h, x, y)]]$. The minimizability gap can be upper-bounded by the approximation error and vanishes when $\mathcal{H} = \mathcal{H}_{\mathrm{all}}$ [Awasthi et al., 2022a,b]. Thus, an $\mathcal{H}$-consistency bound implies Bayes-consistency. The relationship between the two hypothesis-dependent learning guarantees—realizable $\mathcal{H}$-consistency and $\mathcal{H}$-consistency bounds—depends on the target loss adopted in the specific learning scenario. In Section 5, we will demonstrate that in the standard multi-class classification setting, an $\mathcal{H}$-consistency bound is a stronger notion than realizable $\mathcal{H}$-consistency. However, in L2D, these guarantees do not imply one another.

## 3.3 Existing surrogate losses

Here, we will review several consistent surrogate losses used in L2D. For convenience, we use $\widetilde{c}(x, y) = 1_{g(x) \neq y}$ to denote the cost when it specifically represents the expert's classification error, and use $c(x, y)$ when it represents a general cost function.

[Mozannar and Sontag](2020) proposed the first Bayes-consistent surrogate loss by generalizing the cross-entropy loss for L2D, with cost functions based on classification error, which is defined as

$$\mathsf{L}_{\mathrm{CE}}(h, x, y) = -\log\left(\frac{e^{h(x,y)}}{\sum_{y' \in \overline{\mathcal{Y}}} e^{h(x,y')}}\right) - (1 - \widetilde{c}(x,y))\log\left(\frac{e^{h(x,n+1)}}{\sum_{y' \in \overline{\mathcal{Y}}} e^{h(x,y')}}\right).$$

[Verma and Nalisnick](2022) proposed an alternative one-vs-all surrogates loss with cost functions based on expert's classification error, that is Bayes-consistent as well:

$$\mathsf{L}_{\mathrm{OvA}}(h, x, y) = \Phi(h(x,y)) + \sum_{\substack{y' \in \overline{\mathcal{Y}} \\ y' \neq y}} \Phi(-h(x,y')) + (1 - \widetilde{c}(x,y))[\Phi(h(x,n+1)) - \Phi(-h(x,n+1))],$$

where $\Phi$ is a strictly proper binary composite loss [Reid and Williamson, 2010], such as the logistic loss $t \mapsto \log(1 + e^{-t})$. $\mathsf{L}_{\mathrm{CE}}$ and $\mathsf{L}_{\mathrm{OvA}}$ are not realizable $\mathcal{H}$-consistent. Instead, [Mozannar et al.](2023) proposed the following loss function that is realizable $\mathcal{H}$-consistent when $\mathcal{H}$ is closed under scaling:

$$\mathsf{L}_{\mathrm{RS}}(h, x, y) = -2\log\left(\frac{e^{h(x,y)} + (1 - \widetilde{c}(x,y))e^{h(x,n+1)}}{\sum_{y' \in \overline{\mathcal{Y}}} e^{h(x,y')}}\right).$$

However, they were unable to prove or disprove whether the surrogate loss $\mathsf{L}_{\mathrm{RS}}$ is Bayes-consistent.

All the surrogate losses mentioned above and their consistency guarantees hold only for cost functions based on the classification error: $\widetilde{c}(x,y) = 1_{\mathsf{g}(x) \neq y}$. [Mao et al.](2024a) generalized the surrogate loss $\mathsf{L}_{\mathrm{CE}}$ to incorporate general cost functions and any multi-class surrogate losses:

$$\mathsf{L}_{\mathrm{general}}(h, x, y) = \ell(h, x, y) + (1 - c(x,y))\ell(h, x, n+1).$$

Here, $\ell$ is a Bayes-consistent surrogate loss for the multi-class zero-one loss over the augmented label set $\overline{\mathcal{Y}}$. In particular, $\ell$ can be chosen as a comp-sum loss [Mao et al., 2023f], for example, the generalized cross entropy loss (see Section 4.1). As shown by [Mao et al.](2024a), $\mathsf{L}_{\mathrm{general}}$ benefits from $\mathcal{H}$-consistency bounds, which implies its Bayes-consistency.

## 4 Novel surrogate losses

In this section, we introduce a new family of surrogate losses for L2D that benefit from Bayes-consistency, realizable $\mathcal{H}$-consistency and $\mathcal{H}$-consistency bounds, starting from first principles.

### 4.1 Derivation from first principles

Observe that for any $(x,y) \in \mathcal{X} \times \mathcal{Y}$, we have $1_{\mathsf{h}(x)=n+1} = 1_{\mathsf{h}(x) \neq y}1_{\mathsf{h}(x)=n+1}$, since $\mathsf{h}(x) = n+1$ implies $\mathsf{h}(x) \neq y$. Thus, using additionally $1_{\mathsf{h}(x) \in [n]} = 1_{\mathsf{h}(x) \neq n+1}$, the deferral loss can be rewritten as follows for all $(x,y) \in \mathcal{X} \times \mathcal{Y}$:

$$\begin{aligned}
\mathsf{L}_{\mathrm{def}}(h, x, y) &= 1_{\mathsf{h}(x) \neq y}1_{\mathsf{h}(x) \in [n]} + c(x,y)1_{\mathsf{h}(x)=n+1} \\
&= 1_{\mathsf{h}(x) \neq y}1_{\mathsf{h}(x) \neq n+1} + c(x,y)1_{\mathsf{h}(x) \neq y}1_{\mathsf{h}(x)=n+1} \\
&= 1_{\mathsf{h}(x) \neq y}1_{\mathsf{h}(x) \neq n+1} + c(x,y)1_{\mathsf{h}(x) \neq y}(1 - 1_{\mathsf{h}(x) \neq n+1}) \\
&= c(x,y)1_{\mathsf{h}(x) \neq y} + (1 - c(x,y))1_{\mathsf{h}(x) \neq y \wedge \mathsf{h}(x) \neq n+1}. \quad (2)
\end{aligned}$$

Next, we will derive the new surrogate losses for L2D by replacing the indicator functions in (2) with smooth loss functions. The first indicator function $1_{\mathsf{h}(x) \neq y}$ is just the multi-class zero-one loss. Thus, a natural choice is to replace it with a surrogate loss in standard multi-class classification. We will specifically consider the family of comp-sum losses [Mao et al., 2023f], defined as follows for any $(h, x, y) \in \mathcal{H} \times \mathcal{X} \times \mathcal{Y}$:

$$\ell_{\mathrm{comp}}(h, x, y) = \Psi\left(\frac{e^{h(x,y)}}{\sum_{y' \in \overline{\mathcal{Y}}} e^{h(x,y')}}\right),$$

where $\Psi: [0,1] \to \mathbb{R}_+ \cup \{+\infty\}$ is a non-increasing function. For example, by taking $\Psi(t) = -\log(t)$, $\frac{1}{q}(1 - t^q)$ with $q \in (0,1)$, $1 - t$, we obtain the *logistic loss* [Verhulst, 1838, 1845, Berkson, 1944,

Table 1: A new family of surrogate losses $\mathsf{L}_{\mathrm{RL2D}}$ for L2D.

| $\Psi(t)$ | $\mathsf{L}_{\mathrm{RL2D}}$ |
|---|---|
| $-\log(t)$ | $-c(x,y)\log\left[\dfrac{e^{h(x,y)}}{\sum_{y'\in\overline{\mathcal{Y}}}e^{h(x,y')}}\right]-(1-c(x,y))\log\left[\dfrac{e^{h(x,y)}+e^{h(x,n+1)}}{\sum_{y'\in\overline{\mathcal{Y}}}e^{h(x,y')}}\right]$ |
| $\frac{1}{q}(1-t^q)$ | $\dfrac{c(x,y)}{q}\left[1-\left[\dfrac{e^{h(x,y)}}{\sum_{y'\in\overline{\mathcal{Y}}}e^{h(x,y')}}\right]^q\right]+\dfrac{(1-c(x,y))}{q}\left[1-\left[\dfrac{e^{h(x,y)}+e^{h(x,n+1)}}{\sum_{y'\in\overline{\mathcal{Y}}}e^{h(x,y')}}\right]^q\right]$ |
| $1-t$ | $c(x,y)\left(1-\dfrac{e^{h(x,y)}}{\sum_{y'\in\overline{\mathcal{Y}}}e^{h(x,y')}}\right)+(1-c(x,y))\left(1-\dfrac{e^{h(x,y)}+e^{h(x,n+1)}}{\sum_{y'\in\overline{\mathcal{Y}}}e^{h(x,y')}}\right)$ |

1951], the *generalized cross entropy loss* [Zhang and Sabuncu, 2018], and the *mean absolute error loss* [Ghosh et al., 2017], respectively:

Logistic loss:
$$\ell_{\log}(h,x,y)=-\log\left[\frac{e^{h(x,y)}}{\sum_{y'\in\overline{\mathcal{Y}}}e^{h(x,y')}}\right]$$

Generalized cross entropy loss:
$$\ell_{\mathrm{gce}}(h,x,y)=\frac{1}{q}\left[1-\left[\frac{e^{h(x,y)}}{\sum_{y'\in\overline{\mathcal{Y}}}e^{h(x,y')}}\right]^q\right]$$

Mean absolute error loss:
$$\ell_{\mathrm{mae}}(h,x,y)=1-\frac{e^{h(x,y)}}{\sum_{y'\in\overline{\mathcal{Y}}}e^{h(x,y')}}.$$

For any $(h,x,y)\in\mathcal{H}\times\mathcal{X}\times\overline{\mathcal{Y}}$, the confidence margin $\rho_h(x,y)$ is defined by $\rho_h(x,y)=h(x,y)-\max_{y'\in\overline{\mathcal{Y}},y'\neq y}h(x,y')$. Thus, the second indicator function $1_{\mathsf{h}(x)\neq y\wedge\mathsf{h}(x)\neq n+1}$ can be expressed as follows in terms of the confidence margin:

$$1_{\mathsf{h}(x)\neq y\wedge\mathsf{h}(x)\neq n+1}=1_{\left(h(x,y)\leq\max_{y'\in\overline{\mathcal{Y}},y'\neq y}h(x,y')\right)\wedge\left(h(x,n+1)\leq\max_{y'\in\overline{\mathcal{Y}},y'\neq n+1}h(x,y')\right)}$$
$$=1_{(\rho_h(x,y)\leq0)\wedge(\rho_h(x,n+1)\leq0)}$$
$$=1_{\max\{\rho_h(x,y),\rho_h(x,n+1)\}\leq0}.$$

Note that the first indicator function can also be written in terms of margin: $1_{\mathsf{h}(x)\neq y}=1_{\rho_h(x,y)\leq0}$. Unlike the first indicator function, which presses $h(x,y)$ to be the largest score among $\overline{\mathcal{Y}}$, that is the margin $\rho_h(x,y)$ to be positive, the second indicator function only enforces $h(x,y)$ or $h(x,n+1)$ to be the largest score among $\overline{\mathcal{Y}}$, that is the maximum of two margins, $\max\{\rho_h(x,y),\rho_h(x,n+1)\}$, to be positive. This condition can be further strengthened by requiring the sum of two margins, $\rho_h(x,y)+\rho_h(x,n+1)$, to be positive. In view of this observation, we adopt the following modified comp-sum surrogate loss for the second indicator function:

$$\widetilde{\ell}_{\mathrm{comp}}(h,x,y)=\Psi\left(\frac{e^{h(x,y)}+e^{h(x,n+1)}}{\sum_{y'\in\overline{\mathcal{Y}}}e^{h(x,y')}}\right),$$

where $\Psi\colon[0,1]\to\mathbb{R}_+\cup\{+\infty\}$ is a non-increasing function. In other words, $\widetilde{\ell}_{\mathrm{comp}}$ replaces the term $e^{h(x,y)}$ in the softmax function in $\ell_{\mathrm{comp}}$ with the sum $e^{h(x,y)}+e^{h(x,n+1)}$. The effect is to encourage the sum of the two margins, $\rho_h(x,y)+\rho_h(x,n+1)$, to be positive, rather than just the single margin $\rho_h(x,y)$. Following this principle, we derive the following expression for a new family of surrogate losses, $\mathsf{L}_{\mathrm{RL2D}}$, dubbed *realizable L2D*:

$$\mathsf{L}_{\mathrm{RL2D}}(h,x,y)=c(x,y)\ell_{\mathrm{comp}}(h,x,y)+(1-c(x,y))\widetilde{\ell}_{\mathrm{comp}}(h,x,y). \tag{3}$$

For the choices of $\Psi(t)=-\log(t)$, $\frac{1}{q}(1-t^q)$ with $q\in(0,1)$ and $1-t$, we obtain the new surrogate losses for L2D in Table 1. In the next sections, we will prove both realizable $\mathcal{H}$-consistency guarantees and $\mathcal{H}$-consistency bounds for this family of surrogate losses, which imply their excess error bounds and Bayes-consistency as well.

## 4.2 Realizable $\mathcal{H}$-consistency

Here, we show that $\mathsf{L}_{\mathrm{RL2D}}$ is realizable $\mathcal{H}$-consistent with respect to $\mathsf{L}_{\mathrm{def}}$. We say that a hypothesis set $\mathcal{H}$ is *closed under scaling* if $h\in\mathcal{H}\implies\alpha h\in\mathcal{H}$ for any $\alpha\in\mathbb{R}$.

**Theorem 4.1.** *Assume that $\mathcal{H}$ is closed under scaling. Suppose that $\Psi$ is non-increasing, $\Psi\left(\frac{2}{3}\right) > 0$ and $\lim_{t \to 1} \Psi(t) = 0$. Then, the surrogate loss $\mathsf{L}_{\mathrm{RL2D}}$ is realizable $\mathcal{H}$-consistent with respect to $\mathsf{L}_{\mathrm{def}}$.*

The proof, detailed in Appendix A, begins by establishing an upper bound on the deferral loss in terms of the comp-sum loss: $\mathsf{L}_{\mathrm{def}} \leq \frac{\mathsf{L}_{\mathrm{RL2D}}}{\Psi\left(\frac{2}{3}\right)}$. Letting $\hat{h}$ be the minimizer of $\mathsf{L}_{\mathrm{RL2D}}$ and $\alpha$ be any real number, we then show that $\mathcal{E}_{\mathsf{L}_{\mathrm{def}}}(\hat{h}) \leq \frac{1}{\Psi\left(\frac{2}{3}\right)} \mathcal{E}_{\mathsf{L}_{\mathrm{RL2D}}}(\alpha h^{\star})$. The generalization error is then split by conditioning on whether $h^{\star}(x)$ is the deferral class $(n+1)$ or not. Finally, we demonstrate that each conditional term converges to zero as $\alpha$ tends to $+\infty$, and apply the monotone convergence theorem to complete the proof.

### 4.3 $\mathcal{H}$-Consistency bounds

Here, we show that $\mathsf{L}_{\mathrm{RL2D}}$ admits an $\mathcal{H}$-consistency bound with respect to $\mathsf{L}_{\mathrm{def}}$, which implies its Bayes-consistency as well. We say that a hypothesis set is symmetric if there exists a family $\mathcal{F}$ of functions $f$ mapping from $\mathcal{X}$ to $\mathbb{R}$ such that $\{[h(x,1), \ldots, h(x, n+1)] : h \in \mathcal{H}\} = \{[f_1(x), \ldots, f_{n+1}(x)] : f_1, \ldots, f_{n+1} \in \mathcal{F}\}$, for any $x \in \mathcal{X}$. We say that a hypothesis set $\mathcal{H}$ is complete if for any $(x, y) \in \mathcal{X} \times \mathcal{Y}$, the set of scores generated by it spans across the real numbers: $\{h(x, y) \mid h \in \mathcal{H}\} = \mathbb{R}$. Common neural network and linear function hypothesis sets are all symmetric and complete. We first consider the case where the cost is expert's classification error.

**Theorem 4.2.** *Assume that $\mathcal{H}$ is symmetric and complete and that $c(x, y) = 1_{\mathsf{g}(x) \neq y}$. Then, for all $h \in \mathcal{H}$ and any distribution, the following $\mathcal{H}$-consistency bound holds:*

$$\mathcal{E}_{\mathsf{L}_{\mathrm{def}}}(h) - \mathcal{E}_{\mathsf{L}_{\mathrm{def}}}(\mathcal{H}) + \mathcal{M}_{\mathsf{L}_{\mathrm{def}}}(\mathcal{H}) \leq \Gamma(\mathcal{E}_{\mathsf{L}_{\mathrm{RL2D}}}(h) - \mathcal{E}_{\mathsf{L}_{\mathrm{RL2D}}}(\mathcal{H}) + \mathcal{M}_{\mathsf{L}_{\mathrm{RL2D}}}(\mathcal{H})),$$

*where $\Gamma(t) = \sqrt{2t}$ when $\Psi(t) = -\log(t)$ and $\Gamma(t) = \sqrt{2(n+1)^q t}$ when $\Psi(t) = \frac{1}{q}(1 - t^q)$ with $q \in (0, 1)$.*

The proof, detailed in Appendix B.3 and B.4, establishes strong consistency guarantees for our new surrogate loss $\mathsf{L}_{\mathrm{RL2D}}$ (Theorem 4.2). We first introduce $y_{\max} = \operatorname{argmax}_{y \in \mathcal{Y}} p(x, y)$, the label with the highest conditional probability. We then show that for any hypothesis $h$ and input $x$, if $y_{\max}$ is not the predicted label $h_{\max}$, the conditional error of $h$ is lower bounded by a modified hypothesis $\overline{h}$ (obtained by swapping the scores of $y_{\max}$ and $h_{\max}$). Next, for hypotheses where $y_{\max} = h_{\max}$, we lower bound their conditional regret in terms of the conditional regret of the deferral loss using a new hypothesis $h_{\mu}$. This proof is novel and significantly different from existing approaches for establishing $\mathcal{H}$-consistency bounds in either the standard or deferral settings [Mao et al., 2023f, 2024a].

The next result further shows that when $\Psi(t) = 1 - t$, our surrogate losses benefit from $\mathcal{H}$-consistency bounds for any general cost function.

**Theorem 4.3.** *Assume that $\mathcal{H}$ is symmetric and complete. Suppose that $\Psi(t) = 1 - t$. Then, for all $h \in \mathcal{H}$ and any distribution, the following $\mathcal{H}$-consistency bounds hold:*

$$\mathcal{E}_{\mathsf{L}_{\mathrm{def}}}(h) - \mathcal{E}_{\mathsf{L}_{\mathrm{def}}}(\mathcal{H}) + \mathcal{M}_{\mathsf{L}_{\mathrm{def}}}(\mathcal{H}) \leq (n+1)(\mathcal{E}_{\mathsf{L}_{\mathrm{RL2D}}}(h) - \mathcal{E}_{\mathsf{L}_{\mathrm{RL2D}}}(\mathcal{H}) + \mathcal{M}_{\mathsf{L}_{\mathrm{RL2D}}}(\mathcal{H})).$$

The proof is included in Appendix B.2. Theorem 4.2 provides stronger consistency guarantees for our new surrogate loss $\mathsf{L}_{\mathrm{RL2D}}$ with $\Psi(t) = 1 - t$ since it holds for any general cost function. The proof idea is similar to that of Theorem 4.2, albeit with more cases to analyze due to the general cost function. This occurs when lower bounding the conditional regret of a hypothesis $h$, which satisfies $y_{\max} = h_{\max}$, in terms of the conditional regret of the deferral loss by introducing a new hypothesis $h_{\mu}$. The additional cases necessitate a more stringent condition for the guarantee, such that the functions $\Psi(t) = -\log(t)$ and $\Psi(t) = \frac{1}{q}(1 - t^q)$ do not apply.

### 4.4 Excess error bounds and Bayes-consistency

For the family of all measurable functions $\mathcal{H} = \mathcal{H}_{\mathrm{all}}$, the minimizability gaps vanish. In this case, Theorems 4.2 and 4.3 imply the following excess error bounds and Bayes-consistency guarantees.

**Corollary 4.4.** *Suppose that $c(x, y) = 1_{\mathsf{g}(x) \neq y}$. For all $h \in \mathcal{H}_{\mathrm{all}}$ and any distribution, the following excess error bounds hold:*

$$\mathcal{E}_{\mathsf{L}_{\mathrm{def}}}(h) - \mathcal{E}_{\mathsf{L}_{\mathrm{def}}}(\mathcal{H}_{\mathrm{all}}) \leq \Gamma(\mathcal{E}_{\mathsf{L}_{\mathrm{RL2D}}}(h) - \mathcal{E}_{\mathsf{L}_{\mathrm{RL2D}}}(\mathcal{H}_{\mathrm{all}})),$$

Table 2: Consistency properties of existing surrogate losses and ours in the case of $c(x, y) = 1_{\mathbf{g}(x) \neq y}$.

| Surrogate losses | Realizable H-consistency | Bayes-consistency | H-consistency bounds |
|---|---|---|---|
| $\mathsf{L}_{\mathrm{CE}}$ | no | yes | yes |
| $\mathsf{L}_{\mathrm{OvA}}$ | no | yes | yes |
| $\mathsf{L}_{\mathrm{general}}$ | no | yes | yes |
| $\mathsf{L}_{\mathrm{RS}}$ ($\mathsf{L}_{\mathrm{RL2D}}$ with $\Psi(t) = -\log(t)$) | yes | yes (proved by us) | yes (proved by us) |
| $\mathsf{L}_{\mathrm{RL2D}}$ with $\Psi(t) = \frac{1}{q}(1 - t^q), q \in (0, 1)$ | yes | yes | yes |
| $\mathsf{L}_{\mathrm{RL2D}}$ with $\Psi(t) = 1 - t$ | yes | yes | yes |

*where $\Gamma(t) = \sqrt{2t}$ when $\Psi(t) = -\log(t)$ and $\Gamma(t) = \sqrt{2(n + 1)^q t}$ when $\Psi(t) = \frac{1}{q}(1 - t^q)$ with $q \in (0, 1)$. Furthermore, the surrogate loss $\mathsf{L}_{\mathrm{RL2D}}$ is Bayes-consistent with respect to $\mathsf{L}_{\mathrm{def}}$ in these cases.*

**Corollary 4.5.** *Suppose that $\Psi(t) = 1 - t$. For all $h \in \mathcal{H}_{\mathrm{all}}$ and any distribution, the following excess error bounds hold:*

$$\mathcal{E}_{\mathsf{L}_{\mathrm{def}}}(h) - \mathcal{E}_{\mathsf{L}_{\mathrm{def}}}(\mathcal{H}_{\mathrm{all}}) \leq (n + 1)(\mathcal{E}_{\mathsf{L}_{\mathrm{RL2D}}}(h) - \mathcal{E}_{\mathsf{L}_{\mathrm{RL2D}}}(\mathcal{H}_{\mathrm{all}})).$$

*Furthermore, the surrogate loss $\mathsf{L}_{\mathrm{RL2D}}$ is Bayes-consistent with respect to $\mathsf{L}_{\mathrm{def}}$ in this case.*

Therefore, Theorem 4.1 and Corollary 4.4 show that $\mathsf{L}_{\mathrm{RL2D}}$ is both realizable $\mathcal{H}$-consistent and Bayes-consistent with respect to $\mathsf{L}_{\mathrm{def}}$. This solves the open problem raised by Mozannar et al. [2023].

In particular, for cost functions based on classification error, $c(x, y) = 1_{\mathbf{g}(x) \neq y}$, our surrogate loss $\mathsf{L}_{\mathrm{RL2D}}$ with $\Psi(t) = -\log(t)$ coincides with the surrogate loss $\mathsf{L}_{\mathrm{RS}}$ in [Mozannar et al., 2023], modulo a constant. This affirmatively answers the question of whether their surrogate loss is Bayes-consistent when $c(x, y) = 1_{\mathbf{g}(x) \neq y}$. However, their surrogate loss cannot be shown to be Bayes-consistent for a general cost function. In contrast, our surrogate losses $\mathsf{L}_{\mathrm{RL2D}}$ with $\Psi(t) = 1 - t$ are adaptable to general cost functions and benefit from both $\mathcal{H}$-consistency bounds and realizable $\mathcal{H}$-consistency guarantees. We also provide a more general family of comp-sum loss functions with $\Psi(t) = \frac{1}{q}(1 - t^q)$ that benefit from both $\mathcal{H}$-consistency bounds and realizable $\mathcal{H}$-consistency when $c(x, y) = 1_{\mathbf{g}(x) \neq y}$.

### 4.5 Summary

Here, we summarize the consistency properties of existing surrogate losses and ours. As mentioned earlier, most surrogate losses proposed in previous work, except for $\mathsf{L}_{\mathrm{general}}$, are analyzed under the condition $c(x, y) = 1_{\mathbf{g}(x) \neq y}$. This naturally leads to a summary of these surrogate losses in this context, as presented in Table 1. Additionally, we provide analyses and the consistency properties of our surrogate loss, $\mathsf{L}_{\mathrm{RL2D}}$, with general cost functions.

More specifically, our surrogate losses $\mathsf{L}_{\mathrm{RL2D}}$ satisfying Theorem 4.1 perform better in realizable scenarios than the surrogate losses $\mathsf{L}_{\mathrm{CE}}$, $\mathsf{L}_{\mathrm{OVA}}$, and $\mathsf{L}_{\mathrm{general}}$ from prior work, as ours are realizable $\mathcal{H}$-consistent while theirs are not. This will be illustrated by our experiment results in the realizable case (Figure 1a). Our surrogate losses $\mathsf{L}_{\mathrm{RL2D}}$ satisfying Theorem 4.2 and Corollary 4.4 are comparable to the surrogate losses in prior work in non-realizable scenarios when the cost is the expert's classification error, as all of them are Bayes-consistent and supported by H-consistency bounds. This is demonstrated by our experiment in the non-realizable case with the cost function being the expert's classification error (Table 3). Our surrogate losses $\mathsf{L}_{\mathrm{RL2D}}$ satisfying Theorem 4.3 and Corollary 4.5 are superior to the surrogate loss $\mathsf{L}_{\mathrm{RS}}$ in non-realizable scenarios with general cost functions, as ours are supported by H-consistency bounds and Bayes-consistency while theirs are not. This is evidenced by our experiment in the non-realizable case with general cost functions (Figure 1b).

## 5 Relationship between $\mathcal{H}$-consistency bounds and realizable $\mathcal{H}$-consistency

Here, we discuss the relationship between $\mathcal{H}$-consistency bounds and realizable $\mathcal{H}$-consistency. First, realizable $\mathcal{H}$-consistency does not imply $\mathcal{H}$-consistency bounds, since $\mathcal{H}$-consistency bounds require that the relationship holds for all distributions, not just realizable ones. Moreover, $\mathcal{H}$-consistency bounds provide non-asymptotic guarantees, while realizable $\mathcal{H}$-consistency provides only asymptotic guarantees. Second, $\mathcal{H}$-consistency bounds imply realizable $\mathcal{H}$-consistency in the standard multi-class classification setting. This is because minimizability gaps vanish under the realizable assumption in standard case. In particular, for comp-sum losses, the following holds (see Appendix C for proof).

Table 3: Comparison of system accuracy, accepted accuracy and coverage; mean ± standard deviation over three runs. Realizable L2D outperforms or is comparable to baselines in all the settings.

| Method | Dataset | System Accuracy | Accepted Accuracy | Coverage |
|---|---|---|---|---|
| Mozannar and Sontag [2020] ($L_{CE}$) | | $91.60 \pm 0.15$ | $94.61 \pm 0.67$ | $44.55 \pm 1.68$ |
| Verma and Nalisnick [2022] ($L_{OvA}$) | | $92.18 \pm 0.10$ | $95.43 \pm 0.36$ | $58.56 \pm 3.18$ |
| Mozannar et al. [2023] ($L_{RS}$) | HateSpeech | $91.83 \pm 0.63$ | $95.37 \pm 0.72$ | $54.78 \pm 3.70$ |
| Mao et al. [2024a] ($L_{general}$) | | $92.05 \pm 0.04$ | $96.28 \pm 0.35$ | $46.74 \pm 2.80$ |
| Realizable L2D ($L_{RL2D}, q = 0.7$) | | $\underline{92.20 \pm 0.54}$ | $96.06 \pm 0.39$ | $57.85 \pm 0.76$ |
| Realizable L2D ($L_{RL2D}, q = 1$) | | $91.97 \pm 0.29$ | $96.57 \pm 0.69$ | $53.25 \pm 2.49$ |
| Mozannar and Sontag [2020] ($L_{CE}$) | | $66.33 \pm 0.47$ | $73.65 \pm 1.83$ | $55.17 \pm 9.51$ |
| Verma and Nalisnick [2022] ($L_{OvA}$) | | $66.33 \pm 1.31$ | $71.03 \pm 5.10$ | $53.33 \pm 4.73$ |
| Mozannar et al. [2023] ($L_{RS}$) | COMPASS | $66.00 \pm 2.27$ | $63.20 \pm 4.23$ | $69.50 \pm 10.8$ |
| Mao et al. [2024a] ($L_{general}$) | | $66.67 \pm 0.62$ | $76.25 \pm 2.42$ | $48.33 \pm 5.31$ |
| Realizable L2D ($L_{RL2D}, q = 0.7$) | | $66.17 \pm 2.01$ | $69.33 \pm 3.03$ | $55.67 \pm 5.95$ |
| Realizable L2D ($L_{RL2D}, q = 1$) | | $\underline{66.83 \pm 0.85}$ | $69.02 \pm 2.42$ | $54.83 \pm 0.62$ |
| Mozannar and Sontag [2020] ($L_{CE}$) | | $96.27 \pm 0.51$ | $98.77 \pm 0.71$ | $64.33 \pm 6.13$ |
| Verma and Nalisnick [2022] ($L_{OvA}$) | | $96.25 \pm 0.45$ | $98.74 \pm 0.54$ | $67.88 \pm 6.16$ |
| Mozannar et al. [2023] ($L_{RS}$) | CIFAR-10H | $96.63 \pm 0.18$ | $98.23 \pm 0.78$ | $66.63 \pm 1.80$ |
| Mao et al. [2024a] ($L_{general}$) | | $96.75 \pm 0.55$ | $98.65 \pm 0.80$ | $65.68 \pm 3.36$ |
| Realizable L2D ($L_{RL2D}, q = 0.7$) | | $\underline{96.80 \pm 0.25}$ | $98.37 \pm 0.20$ | $76.77 \pm 3.63$ |
| Realizable L2D ($L_{RL2D}, q = 1$) | | $96.57 \pm 0.05$ | $98.34 \pm 0.24$ | $77.37 \pm 2.43$ |

**Theorem 5.1.** *Assume that there exists a zero error solution $h^\star \in \mathcal{H}$ with $\mathcal{E}_{\ell_{0-1}}(h^\star) = 0$ and $\mathcal{H}$ is closed under scaling. Assume that $\lim_{t \to 1} \Psi(t) = 0$. Then, the minimizability gap of comp-sum loss $\ell_{comp}$ vanishes: $\mathcal{M}_{\ell_{comp}}(\mathcal{H}) = 0$.*

However, in the deferral setting, this relationship no longer holds: $\mathcal{H}$-consistency bounds cannot imply realizable $\mathcal{H}$-consistency. In particular, Mao et al. [2023f] showed that $L_{CE}$ benefits from $\mathcal{H}$-consistency bounds, while Mozannar et al. [2023] showed that it is not realizable $\mathcal{H}$-consistent. The loss function in [Madras et al., 2018] is not Bayes-consistent, and thus does not have $\mathcal{H}$-consistency bound guarantees, but is actually realizable $\mathcal{H}$-consistent [Mozannar et al., 2023].

# 6 Experiments

In this section, we empirically evaluate our proposed surrogate losses and compare them with existing baselines.

**Experimental settings.** We follow the setting of Mozannar et al. [2023] and conduct experiments on a synthetic dataset: Mixture-of-Gaussians [Mozannar et al., 2023], and three real-world datasets: CIFAR-10H [Battleday et al., 2020], HateSpeech [Davidson et al., 2017], and COMPASS [Dressel and Farid, 2018]. For these three datasets, we adopt the same model class as that in [Mozannar et al., 2023, Table 1]. Each dataset is randomly split into 70%, 10%, and 20% for training, validation, and testing, respectively. For the Mixture-of-Gaussians, we adopt the exact realizable setting from [Mozannar et al., 2023, Section 7.2], which is realizable by linear functions: there exists a linear hypothesis $h^\star \in \mathcal{H}$ achieving zero deferral loss, $\mathcal{E}_{L_{def}}(h^\star) = 0$.

As with [Mozannar et al., 2023], we choose the cost function to be the expert's classification error: $c(x, y) = 1_{g(x) \neq y}$. We compare our surrogate to four baselines as described in Section 3.3: the cross-entropy surrogate $L_{CE}$ from [Mozannar and Sontag, 2020], the one-vs-all surrogate $L_{OvA}$ from [Mozannar and Sontag, 2020], the realizable surrogate $L_{RS}$ from [Mozannar et al., 2023], and the general surrogate $L_{general}$ from [Mao et al., 2024a]. For $L_{OvA}$, we choose $\Phi$ as the logistic loss, following [Verma and Nalisnick, 2022]. For $L_{general}$, we choose $\ell$ as the generalized cross entropy loss with $q = 0.7$, following [Mao et al., 2024a]. For our Realizable L2D surrogate $L_{RL2D}$, we consider two choices: $\ell$ as the generalized cross entropy loss with $q = 0.7$, following [Zhang and Sabuncu, 2018, Mao et al., 2024a], and $\ell$ as the mean absolute error loss ($q = 1$). Among these, $L_{CE}$, $L_{OvA}$ and $L_{general}$ are Bayes-consistent but not realizable $\mathcal{H}$-consistent; $L_{RS}$, $L_{RL2D}$ with $q = 0.7$ and $L_{RL2D}$ with $q = 1$ are both Bayes-consistent and realizable $\mathcal{H}$-consistent, as shown in Sections 4.2 and 4.4. Note that in this case, $L_{RS}$ is a special case of $L_{RL2D}$ when $\Psi$ is chosen as $t \mapsto -\log(t)$. We use the same optimizer, learning rate, and number of epochs as chosen in [Mozannar et al., 2023], and we select the model that achieves the highest *system accuracy*, that is average $[1 - L_{def}(h, x, y)]$, on a validation set.

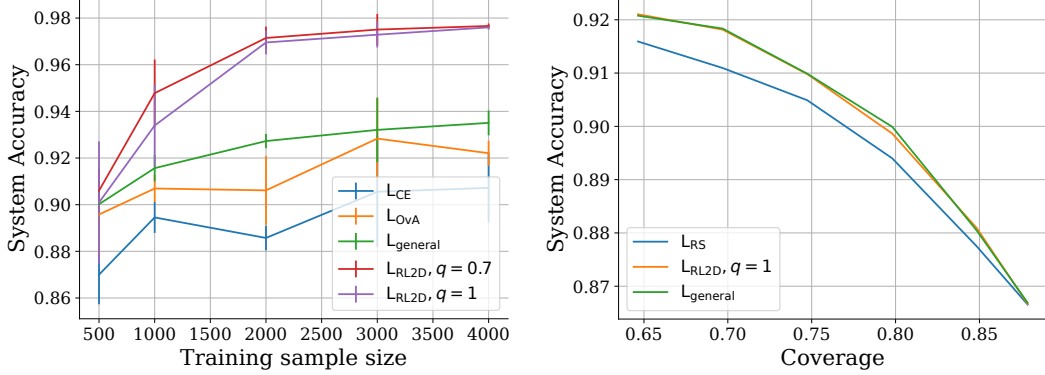

(a) Comparison of system accuracy versus training sample size on a realizable synthetic dataset.

(b) Comparison of system accuracy versus coverage on the HateSpeech dataset with general cost functions.

Figure 1: Results for the realizable case and the non-realizable case with general cost functions.

**Evaluation.** For the three real-world datasets, we report the *system accuracy*, that is average value of $[1 - \mathsf{L}_{\mathrm{def}}(h, x, y)]$ on the test data. For completeness, we also include the *accepted accuracy*, that is the average value of $[1_{\mathsf{h}(x) \neq y} 1_{\mathsf{h}(x) \in [n]}]$. This metric considers only incorrect predictions ($\mathsf{h}(x) \neq y$) and measures the fraction of those where the system's output ($\mathsf{h}(x)$) falls within the valid range of possible outputs ($[n]$). We also report the *coverage*, that is the average value of $[1_{\mathsf{h}(x) \in [n]}]$ on the test set, or the fraction of test instances where the system's prediction falls within the valid range ($[n]$). For each metric, we average results over three runs and report the mean accuracy along with the standard deviation for both our proposed methods and the baseline approaches. For the realizable Mixture-of-Gaussians, we plot the system accuracy of various methods on a held-out test dataset consisting of 5,000 points as we increase the size of the training data.

**Results.** Table 3 shows that for the real-world datasets, $\mathsf{L}_{\mathrm{RL2D}}$ with $q = 0.7$, and $\mathsf{L}_{\mathrm{RL2D}}$ with $q = 1$ either outperform or are comparable to the best baseline in terms of system accuracy on each dataset. This performance is supported by our $\mathcal{H}$-consistency bounds and Bayes-consistency results for our Realizable L2D surrogate with respect to the deferral loss $\mathsf{L}_{\mathrm{def}}$, as shown in Sections 4.3 and 4.4. Table 3 also shows that $\mathsf{L}_{\mathrm{RL2D}}$ achieves reasonable coverage and acceptable accuracy. The system accuracy, coverage, and standard deviations of the baselines match those in [Mozannar et al., 2023]. Moreover, $\mathsf{L}_{\mathrm{RS}}$, $\mathsf{L}_{\mathrm{RL2D}}$ with $q = 0.7$, and $\mathsf{L}_{\mathrm{RL2D}}$ with $q = 1$ perform differently across various datasets: $\mathsf{L}_{\mathrm{RL2D}}$ with $q = 0.7$ outperforms the others on HateSpeech and CIFAR-10H, while $\mathsf{L}_{\mathrm{RL2D}}$ with $q = 1$ outperforms the others on COMPASS. Note that in this case, $\mathsf{L}_{\mathrm{RS}}$ is a special case of $\mathsf{L}_{\mathrm{RL2D}}$ when $\Psi$ is chosen as $t \mapsto -\log(t)$. These results show that Realizable L2D can benefit from the flexibility in the choice of $\Psi$.

Figure 1a shows system accuracy versus training samples on the realizable Mixture-of-Gaussians distribution. Our surrogate loss $\mathsf{L}_{\mathrm{RL2D}}$ with $q = 0.7$ and $q = 1$ are realizable $\mathcal{H}$-consistent, while $\mathsf{L}_{\mathrm{CE}}$, $\mathsf{L}_{\mathrm{OVA}}$ and $\mathsf{L}_{\mathrm{general}}$ are not. This verifies our theory.

Figure 1b shows system accuracy versus coverage on the HateSpeech dataset by varying $\beta$ in the general cost functions $c(x, y) = 1_{\mathsf{g}(x) \neq y} + \beta$. As $\beta$ increases, deferral algorithms yield solutions with higher coverage and decreased system accuracy. This is because $\beta$ controls the trade-off between expert's inference cost and accuracy. $\mathsf{L}_{\mathrm{RL2D}}$ with $q = 1$ performs comparably to the surrogate loss $\mathsf{L}_{\mathrm{general}}$, as both are supported by $\mathcal{H}$-consistency bounds and Bayes-consistency with general cost functions. Our surrogate loss $\mathsf{L}_{\mathrm{RL2D}}$ with $q = 1$ outperforms $\mathsf{L}_{\mathrm{RS}}$ because the latter does not benefit from Bayes-consistency with general cost functions.

# 7 Conclusion

We introduced a broad family of surrogate losses and algorithms for learning to defer, parameterized by a non-increasing function. We established their realizable $\mathcal{H}$-consistency properties under mild conditions and proved that several of these surrogate losses benefit from $\mathcal{H}$-consistency bounds for cost functions based on classification error and general cost functions, which also imply their Bayes-consistency. This research not only resolves an open question posed in previous work but also lays the groundwork for comparing various consistency notions in learning to defer and standard classification. Looking forward, our approach offers a promising avenue for analyzing multi-expert and two-stage settings.

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

# Contents of Appendix

# A   Proof of realizable $\mathcal{H}$-consistency

**Theorem 4.1.** *Assume that $\mathcal{H}$ is closed under scaling. Suppose that $\Psi$ is non-increasing, $\Psi\left(\frac{2}{3}\right) > 0$ and $\lim_{t \to 1} \Psi(t) = 0$. Then, the surrogate loss $\mathsf{L}_{\mathrm{RL2D}}$ is realizable $\mathcal{H}$-consistent with respect to $\mathsf{L}_{\mathrm{def}}$.*

*Proof.* We first prove that for every $(h, x, y) \in \mathcal{H} \times \mathcal{X} \times \mathcal{Y}$, the following inequality holds:

$$\mathsf{L}_{\mathrm{def}}(h, x, y) \le \frac{\mathsf{L}_{\mathrm{RL2D}}(h, x, y)}{\Psi\left(\frac{2}{3}\right)}.$$

We will analyze case by case.

1. **Case I**: If $\mathsf{h}(x) \in [n]$ (deferral does not occur):

   (a) If $\mathbb{1}_{\mathsf{h}(x) \ne y} = 1$, then we must have

   $$\mathsf{L}_{\mathrm{def}}(h, x, y) = 1, \quad \frac{e^{h(x,y)}}{\sum_{y' \in \overline{\mathcal{Y}}} e^{h(x,y')}} \le \frac{1}{2}, \quad \frac{e^{h(x,y)} + e^{h(x,n+1)}}{\sum_{y' \in \overline{\mathcal{Y}}} e^{h(x,y')}} \le \frac{2}{3}$$

   $$\implies \mathsf{L}_{\mathrm{RL2D}}(h, x, y) \ge c(x, y) \Psi\left(\frac{1}{2}\right) + (1 - c(x, y)) \Psi\left(\frac{2}{3}\right) \ge \Psi\left(\frac{2}{3}\right) \mathsf{L}_{\mathrm{def}}(h, x, y).$$

   (b) If $\mathbb{1}_{\mathsf{h}(x) \ne y} = 0$, then we must have

   $$\mathsf{L}_{\mathrm{RL2D}}(h, x, y) \ge 0 = \mathsf{L}_{\mathrm{def}}(h, x, y).$$

2. **Case II**: If $\mathsf{h}(x) = n + 1$ (deferral occurs): then we must have

   $$\mathsf{L}_{\mathrm{def}}(h, x, y) = c(x, y), \quad \frac{e^{h(x,y)}}{\sum_{y' \in \overline{\mathcal{Y}}} e^{h(x,y')}} \le \frac{1}{2}$$

   $$\implies \mathsf{L}_{\mathrm{RL2D}}(h, x, y) \ge c(x, y) \Psi\left(\frac{1}{2}\right) \ge \Psi\left(\frac{2}{3}\right) \mathsf{L}_{\mathrm{def}}(h, x, y).$$

This concludes that $\mathsf{L}_{\mathrm{def}}(h, x, y) \le \frac{\mathsf{L}_{\mathrm{RL2D}}(h,x,y)}{\Psi\left(\frac{2}{3}\right)}$. Next, we prove that $\mathsf{L}_{\mathrm{RL2D}}$ is realizable $\mathcal{H}$-consistent under the assumptions. Consider a distribution and an expert under which there exists a zero error solution $h^* \in \mathcal{H}$ with $\mathcal{E}_{\mathsf{L}_{\mathrm{def}}}(h^*) = 0$. Let $\hat{h}$ be the minimizer of the surrogate loss: $\hat{h} \in \operatorname{argmin}_{h \in \mathcal{H}} \mathcal{E}_{\mathsf{L}_{\mathrm{RL2D}}}(h)$. Let $\alpha$ be any real number. Then, the following inequality holds:

$$\mathcal{E}_{\mathsf{L}_{\mathrm{def}}}(\hat{h}) \le \frac{1}{\Psi\left(\frac{2}{3}\right)} \mathcal{E}_{\mathsf{L}_{\mathrm{RL2D}}}(\hat{h}) \qquad\qquad (\mathsf{L}_{\mathrm{def}} \le \tfrac{1}{\Psi\left(\frac{2}{3}\right)} \mathsf{L}_{\mathrm{RL2D}})$$

$$\le \frac{1}{\Psi\left(\frac{2}{3}\right)} \mathcal{E}_{\mathsf{L}_{\mathrm{RL2D}}}(\alpha h^*) \qquad (\hat{h} \in \operatorname{argmin}_{h \in \mathcal{H}} \mathcal{E}_{\mathsf{L}_{\mathrm{RL2D}}}(h) \text{ and } \mathcal{H} \text{ is closed under scaling})$$

$$= \frac{1}{\Psi\left(\frac{2}{3}\right)} \mathbb{E}[\mathsf{L}_{\mathrm{RL2D}}(\alpha h^*, x, y) \mid \mathsf{h}^*(x) = n + 1] \mathbb{P}(\mathsf{h}^*(x) = n + 1)$$

$$+ \frac{1}{\Psi\left(\frac{2}{3}\right)} \mathbb{E}[\mathsf{L}_{\mathrm{RL2D}}(\alpha h^*, x, y) \mid \mathsf{h}^*(x) \in [n]] \mathbb{P}(\mathsf{h}^*(x) \in [n]).$$

For the first term conditional on $\mathsf{h}^*(x) = n + 1$, we must have $h^*(x, n + 1) > \max_{y \in \mathcal{Y}} h^*(x, y)$ and $c(x, y) = 0$ since the data is realizable. Therefore,

$$\lim_{\alpha \to +\infty} \mathbb{E}[\mathsf{L}_{\mathrm{RL2D}}(\alpha h^*, x, y) \mid \mathsf{h}^*(x) = n + 1] \mathbb{P}(\mathsf{h}^*(x) = n + 1)$$

$$= \lim_{\alpha \to +\infty} \mathbb{E}\left[\Psi\left(\frac{e^{\alpha h^*(x,y)} + e^{\alpha h^*(x,n+1)}}{\sum_{y' \in \overline{\mathcal{Y}}} e^{\alpha h^*(x,y')}}\right) \;\middle|\; \mathsf{h}^*(x) = n + 1\right] \mathbb{P}(\mathsf{h}^*(x) = n + 1)$$

$$= \mathbb{E}[0 \mid \mathsf{h}^*(x) = n + 1] \mathbb{P}(\mathsf{h}^*(x) = n + 1) \quad (\lim_{t \to 1} \Psi(t) = 0 \text{ and monotone convergence theorem})$$

$$= 0.$$

For the second term conditional on $h^*(x) \in [n]$, we must have $h^*(x,y) > \max_{y' \in \overline{y}, y' \neq y} h(x,y')$ since the data is realizable. Therefore,

$$\lim_{\alpha \to +\infty} \mathbb{E}[\mathsf{L}_{\mathrm{RL2D}}(\alpha h^*, x, y) \mid h^*(x) \in [n]]\mathbb{P}(h^*(x) \in [n])$$

$$= \lim_{\alpha \to +\infty} \mathbb{E}\Bigg[ c(x,y)\Psi\Bigg( \frac{e^{\alpha h^*(x,y)}}{\sum_{y' \in \overline{y}} e^{\alpha h^*(x,y')}} \Bigg)$$

$$+ (1 - c(x,y))\Psi\Bigg( \frac{e^{\alpha h^*(x,y)} + e^{\alpha h^*(x,n+1)}}{\sum_{y' \in \overline{y}} e^{\alpha h^*(x,y')}} \Bigg) \mid h^*(x) \in [n] \Bigg]\mathbb{P}(h^*(x) \in [n])$$

$$= \mathbb{E}[0 \mid h^*(x) \in [n]]\mathbb{P}(h^*(x) \in [n]) \qquad (\lim_{t \to 1} \Psi(t) = 0 \text{ and monotone convergence theorem})$$

$$= 0.$$

Combining the two analyses, we conclude that $\mathcal{E}_{\mathsf{L}_{\mathrm{def}}}(\hat{h}) = 0$ and thus $\mathsf{L}_{\mathrm{RL2D}}$ is realizable $\mathcal{H}$-consistent with respect to $\mathsf{L}_{\mathrm{def}}$. □

# B  Proof of $\mathcal{H}$-consistency bounds

Before delving into the proof, we first establish some essential notation and definitions. Let $\mathsf{L}$ represent a deferral surrogate loss and $\mathcal{H}$ denote a hypothesis set. We define the conditional error as $\mathcal{C}_{\mathsf{L}}(h,x) = \mathbb{E}_{y|x}[\mathsf{L}(h,x,y)]$, the best-in-class conditional error as $\mathcal{C}_{\mathsf{L}}^*(\mathcal{H},x) = \inf_{h \in \mathcal{H}} \mathcal{C}_{\mathsf{L}}(h,x)$, and the conditional regret as $\Delta\mathcal{C}_{\mathsf{L},\mathcal{H}}(h,x) = \mathcal{C}_{\mathsf{L}}(h,x) - \mathcal{C}_{\mathsf{L}}^*(\mathcal{H},x)$. We proceed to present a general theorem demonstrating that, to establish $\mathcal{H}$-consistency bounds (1) with a concave function $\Gamma$, it suffices to lower bound the conditional regret of the surrogate loss by that of the deferral loss, using the same function $\Gamma$.

**Theorem B.1.** *If the following holds for all $h \in \mathcal{H}$ and $x \in \mathcal{X}$, for some concave function $\Gamma$:*

$$\Delta\mathcal{C}_{\mathsf{L}_{\mathrm{def}},\mathcal{H}}(h,x) \leq \Gamma(\Delta\mathcal{C}_{\mathsf{L},\mathcal{H}}(h,x)), \tag{4}$$

*then, for all hypotheses $h \in \mathcal{H}$ and for any distribution,*

$$\mathcal{E}_{\mathsf{L}_{\mathrm{def}}}(h) - \mathcal{E}_{\mathsf{L}_{\mathrm{def}}}^*(\mathcal{H}) + \mathcal{M}_{\mathsf{L}_{\mathrm{def}}}(\mathcal{H}) \leq \Gamma(\mathcal{E}_{\mathsf{L}}(h) - \mathcal{E}_{\mathsf{L}}^*(\mathcal{H}) + \mathcal{M}_{\mathsf{L}}(\mathcal{H})).$$

*Proof.* We can express the expectations of the conditional regrets for $\mathsf{L}_{\mathrm{def}}$ and $\mathsf{L}$ as follows:

$$\mathbb{E}_x[\Delta\mathcal{C}_{\mathsf{L}_{\mathrm{def}},\mathcal{H}}(h,x)] = \mathcal{E}_{\mathsf{L}_{\mathrm{def}}}(h) - \mathcal{E}_{\mathsf{L}_{\mathrm{def}}}^*(\mathcal{H}) + \mathcal{M}_{\mathsf{L}_{\mathrm{def}}}(\mathcal{H})$$

$$\mathbb{E}_x[\Delta\mathcal{C}_{\mathsf{L},\mathcal{H}}(h,x)] = \mathcal{E}_{\mathsf{L}}(h) - \mathcal{E}_{\mathsf{L}}^*(\mathcal{H}) + \mathcal{M}_{\mathsf{L}}(\mathcal{H}).$$

Then, by using (4) and taking the expectation, we obtain:

$$\mathcal{E}_{\mathsf{L}_{\mathrm{def}}}(h) - \mathcal{E}_{\mathsf{L}_{\mathrm{def}}}^*(\mathcal{H}) + \mathcal{M}_{\mathsf{L}_{\mathrm{def}}}(\mathcal{H}) = \mathbb{E}_x[\Delta\mathcal{C}_{\mathsf{L}_{\mathrm{def}},\mathcal{H}}(h,x)]$$

$$\leq \mathbb{E}_x[\Gamma(\Delta\mathcal{C}_{\mathsf{L},\mathcal{H}}(h,x))] \qquad\qquad \text{(Eq. (4))}$$

$$\leq \Gamma\Big(\mathbb{E}_x[\Delta\mathcal{C}_{\mathsf{L},\mathcal{H}}(h,x)]\Big) \qquad\qquad \text{(concavity of } \Gamma)$$

$$= \Gamma(\mathcal{E}_{\mathsf{L}}(h) - \mathcal{E}_{\mathsf{L}}^*(\mathcal{H}) + \mathcal{M}_{\mathsf{L}}(\mathcal{H})).$$

Thus, the proof is complete. □

Next, to prove $\mathcal{H}$-consistency bounds using Theorem B.1, we will characterize the conditional regret of the deferral loss $\mathsf{L}_{\mathrm{def}}$ in the following section.

## B.1  Auxiliary lemma

To simplify the presentation, we introduce the following notation. For any $y \in \mathcal{Y}$, define $p(x,y) = \mathbb{P}(Y = y \mid X = x)$ as the conditional probability that $Y = y$ given $X = x$. For brevity, we will omit the dependency on $x$ in our notation. We denote by $h_y = h(x,y)$ for any $y \in \overline{\mathcal{Y}}$. We also denote by $p_y = p(x,y)$ and $q_y = p(x,y)c(x,y)$ for any $y \in \mathcal{Y}$, and $p_{n+1} = \sum_{y \in \mathcal{Y}} p(x,y)(1 - c(x,y))$.

Note that $p(x,y)(1-c(x,y)) = p_y - q_y,\ \forall y \in \mathcal{Y}$. Let $p_{\mathsf{h}} = p_{\mathsf{h}(x)} = \begin{cases} p_{\mathsf{h}(x)} & \mathsf{h}(x) \in [n] \\ p_{n+1} & \mathsf{h}(x) = n+1. \end{cases}$ Let $y_{\max} = \operatorname{argmax}_{y \in \mathcal{Y}} p_y$ and $h_{\max} = \operatorname{argmax}_{y \in \mathcal{Y}} h_y$. Note that both $y_{\max}$ and $h_{\max}$ are in the label space $\mathcal{Y}$, while $\mathsf{h}(x)$ is in the augmented label space $\overline{\mathcal{Y}}$. We characterize the conditional regret of the deferral loss $\mathsf{L}_{\text{def}}$ as follows.

**Lemma B.2.** *Assume that $\mathcal{H}$ is symmetric and complete. Then, the conditional regret of the deferral loss $\mathsf{L}_{\text{def}}$ can be expressed as follows:* $\Delta\mathcal{C}_{\mathsf{L}_{\text{def}},\mathcal{H}}(h,x) = \max\{p_{y_{\max}}, p_{n+1}\} - p_{\mathsf{h}}$.

*Proof.* We can write the conditional error of the deferral loss as follows:

$$
\begin{aligned}
&\mathcal{C}_{\mathsf{L}_{\text{def}}}(h,x) \\
&= \sum_{y \in \mathcal{Y}} p(x,y) \mathsf{L}_{\text{def}}(h,x,y) \\
&= \sum_{y \in \mathcal{Y}} p(x,y) 1_{\mathsf{h}(x) \neq y} 1_{\mathsf{h}(x) \in [n]} + \sum_{y \in \mathcal{Y}} p(x,y) c(x,y) 1_{\mathsf{h}(x)=n+1} \\
&= (1 - p_{\mathsf{h}(x)}) 1_{\mathsf{h}(x) \in [n]} + (1 - p_{n+1}) 1_{\mathsf{h}(x)=n+1} \\
&= 1 - p_{\mathsf{h}}.
\end{aligned}
$$

Since $\mathcal{H}$ is symmetric and complete, for any $x \in \mathcal{X}$, $\{\mathsf{h}(x) : h \in \mathcal{H}\} = \overline{\mathcal{Y}}$. Then, the best-in-class conditional error of $\mathsf{L}_{\text{def}}$ can be expressed as follows:

$$
\mathcal{C}^*_{\mathsf{L}_{\text{def}}}(\mathcal{H},x) = \inf_{h \in \mathcal{H}} \mathcal{C}_{\mathsf{L}_{\text{def}}}(h,x) = 1 - \max\{p_{n+1}, p_{y_{\max}}\} \tag{5}
$$

Therefore, $\Delta\mathcal{C}_{\mathsf{L}_{\text{def}},\mathcal{H}}(h,x) = \mathcal{C}_{\mathsf{L}_{\text{def}}}(h,x) - \mathcal{C}^*_{\mathsf{L}_{\text{def}}}(\mathcal{H},x) = \max\{p_{y_{\max}}, p_{n+1}\} - p_{\mathsf{h}}$. $\qquad\square$

Next, we will present the proofs separately in the following sections, by lower bounding the conditional regret of the surrogate loss $\mathsf{L}$ by that of the deferral loss $\mathsf{L}_{\text{def}}$ using Lemma B.2.

**B.2** $\Psi(t) = 1 - t$

**Theorem B.3.** *Assume that $\mathcal{H}$ is symmetric and complete. Then, for all $h \in \mathcal{H}$ and any distribution, the following $\mathcal{H}$-consistency bound holds:*

$$
\mathcal{E}_{\mathsf{L}_{\text{def}}}(h) - \mathcal{E}_{\mathsf{L}_{\text{def}}}(\mathcal{H}) + \mathcal{M}_{\mathsf{L}_{\text{def}}}(\mathcal{H}) \leq n\left(\mathcal{E}_{\mathsf{L}_{\text{RL2D}}}(h) - \mathcal{E}_{\mathsf{L}_{\text{RL2D}}}(\mathcal{H}) + \mathcal{M}_{\mathsf{L}_{\text{RL2D}}}(\mathcal{H})\right).
$$

*Proof.* We can write the conditional error of the surrogate loss as follows:

$$
\begin{aligned}
&\mathcal{C}_{\mathsf{L}_{\text{RL2D}}}(h,x) \\
&= \sum_{y \in \mathcal{Y}} p(x,y) \mathsf{L}_{\text{RL2D}}(h,x,y) \\
&= \sum_{y \in \mathcal{Y}} p(x,y) c(x,y) \left(1 - \frac{e^{h(x,y)}}{\sum_{y' \in \overline{\mathcal{Y}}} e^{h(x,y')}}\right) + \sum_{y \in \mathcal{Y}} p(x,y)(1 - c(x,y)) \left(1 - \frac{e^{h(x,y)} + e^{h(x,n+1)}}{\sum_{y' \in \overline{\mathcal{Y}}} e^{h(x,y')}}\right) \\
&= \sum_{y \in \mathcal{Y}} q_y \left(1 - \frac{e^{h_y}}{\sum_{y' \in \overline{\mathcal{Y}}} e^{h_{y'}}}\right) + \sum_{y \in \mathcal{Y}} (p_y - q_y) \left(1 - \frac{e^{h_y} + e^{h_{n+1}}}{\sum_{y' \in \overline{\mathcal{Y}}} e^{h_{y'}}}\right).
\end{aligned}
$$

By Lemma B.2, the conditional regret of the deferral loss can be expressed as

$$
\Delta\mathcal{C}_{\mathsf{L}_{\text{def}},\mathcal{H}}(h,x) = \max\{p_{y_{\max}}, p_{n+1}\} - p_{\mathsf{h}}.
$$

Next, we will show that the conditional regret of the surrogate loss can be lower bounded as follows:

$$
\Delta\mathcal{C}_{\mathsf{L}_{\text{RL2D}},\mathcal{H}}(h,x) = \mathcal{C}_{\mathsf{L}_{\text{RL2D}}}(h) - \mathcal{C}^*_{\mathsf{L}_{\text{RL2D}}}(\mathcal{H}) \geq \frac{1}{n+1}\left(\Delta\mathcal{C}_{\mathsf{L}_{\text{def}},\mathcal{H}}(h,x)\right). \tag{6}
$$

We first prove that for any hypothesis $h$ and $x \in \mathcal{X}$, if $y_{\max} \neq h_{\max}$, then the conditional error of $h$ can be lower bounded by that of $\overline{h}$, which satisfies that $\overline{h}(x, y) = \begin{cases} h_{h_{\max}} & y = y_{\max} \\ h_{y_{\max}} & y = h_{\max} \\ h_y & \text{otherwise.} \end{cases}$ . Indeed,

$$
\mathcal{C}_{\mathsf{L_{RL2D}}}(h) - \mathcal{C}_{\mathsf{L_{RL2D}}}(\overline{h}) = q_{y_{\max}}\left(1 - \frac{e^{h_{y_{\max}}}}{\sum_{y' \in \overline{y}} e^{h_{y'}}}\right) + (p_{y_{\max}} - q_{y_{\max}})\left(1 - \frac{e^{h_{y_{\max}}} + e^{h_{n+1}}}{\sum_{y' \in \overline{y}} e^{h_{y'}}}\right)
$$

$$
+ q_{h_{\max}}\left(1 - \frac{e^{h_{h_{\max}}}}{\sum_{y' \in \overline{y}} e^{h_{y'}}}\right) + (p_{h_{\max}} - q_{h_{\max}})\left(1 - \frac{e^{h_{h_{\max}}} + e^{h_{n+1}}}{\sum_{y' \in \overline{y}} e^{h_{y'}}}\right)
$$

$$
- q_{y_{\max}}\left(1 - \frac{e^{h_{h_{\max}}}}{\sum_{y' \in \overline{y}} e^{h_{y'}}}\right) - (p_{y_{\max}} - q_{y_{\max}})\left(1 - \frac{e^{h_{h_{\max}}} + e^{h_{n+1}}}{\sum_{y' \in \overline{y}} e^{h_{y'}}}\right)
$$

$$
- q_{h_{\max}}\left(1 - \frac{e^{h_{y_{\max}}}}{\sum_{y' \in \overline{y}} e^{h_{y'}}}\right) - (p_{h_{\max}} - q_{h_{\max}})\left(1 - \frac{e^{h_{y_{\max}}} + e^{h_{n+1}}}{\sum_{y' \in \overline{y}} e^{h_{y'}}}\right)
$$

$$
= \frac{1}{\sum_{y' \in \overline{y}} e^{h_{y'}}}(p_{y_{\max}} - p_{h_{\max}})\left(e^{h_{h_{\max}}} - e^{h_{y_{\max}}}\right) \geq 0.
$$

Therefore, we only need to lower bound the conditional regret of hypothesis $h$ satisfying $y_{\max} = h_{\max}$. Next, we will analyze case by case. Note that when $(p_{y_{\max}} - p_{n+1})(h_{y_{\max}} - h_{n+1}) > 0$, we have $\Delta \mathcal{C}_{\mathsf{L_{def}}, \mathcal{H}}(h, x) = \max\{p_{y_{\max}}, p_{n+1}\} - p_{\mathsf{h}} = 0$.

1. **Case I**: If $p_{y_{\max}} - p_{n+1} \geq 0$ and $h_{y_{\max}} - h_{n+1} \leq 0$: we define a new hypothesis $h_\mu$ such that $h_\mu(x, y) = \begin{cases} \log(e^{h_{n+1}} + \mu) & y = y_{\max} \\ \log(e^{h_{y_{\max}}} - \mu) & y = n + 1 \\ h(x, y) & \text{otherwise.} \end{cases}$ , where $e^{h_{y_{\max}}} \geq \mu \geq 0$. Then, we can lower bound the conditional regret of $\mathsf{L_{RL2D}}$ by using $\Delta \mathcal{C}_{\mathsf{L_{RL2D}}, \mathcal{H}}(h, x) \geq \mathcal{C}_{\mathsf{L_{RL2D}}}(h) - \mathcal{C}^*_{\mathsf{L_{RL2D}}}(h_\mu)$ for any $e^{h_{y_{\max}}} \geq \mu \geq 0$:

$$
\Delta \mathcal{C}_{\mathsf{L_{RL2D}}, \mathcal{H}}(h, x)
$$

$$
\geq \sup_{e^{h_{y_{\max}}} \geq \mu \geq 0} \left(\mathcal{C}_{\mathsf{L_{RL2D}}}(h) - \mathcal{C}^*_{\mathsf{L_{RL2D}}}(h_\mu)\right)
$$

$$
\geq \sup_{e^{h_{y_{\max}}} \geq \mu \geq 0} \left(q_{y_{\max}}\left(1 - \frac{e^{h_{y_{\max}}}}{\sum_{y' \in \overline{y}} e^{h_{y'}}}\right) + (p_{y_{\max}} - q_{y_{\max}})\left(1 - \frac{e^{h_{y_{\max}}} + e^{h_{n+1}}}{\sum_{y' \in \overline{y}} e^{h_{y'}}}\right)\right.
$$

$$
+ \sum_{y' \in \mathcal{Y}, y' \neq y_{\max}} (p_{y'} - q_{y'})\left(1 - \frac{e^{h_{y'}} + e^{h_{n+1}}}{\sum_{y' \in \overline{y}} e^{h_{y'}}}\right)
$$

$$
- q_{y_{\max}}\left(1 - \frac{e^{h_{n+1}} + \mu}{\sum_{y' \in \overline{y}} e^{h_{y'}}}\right) - (p_{y_{\max}} - q_{y_{\max}})\left(1 - \frac{e^{h_{n+1}} + e^{h_{h_{\max}}}}{\sum_{y' \in \overline{y}} e^{h_{y'}}}\right)\right)
$$

$$
- \sum_{y' \in \mathcal{Y}, y' \neq y_{\max}} (p_{y'} - q_{y'})\left(1 - \frac{e^{h_{y'}} + e^{h_{y_{\max}}} - \mu}{\sum_{y' \in \overline{y}} e^{h_{y'}}}\right)
$$

$$
= \frac{1}{\sum_{y' \in \overline{y}} e^{h_{y'}}} \sup_{e^{h_{y_{\max}}} \geq \mu \geq 0} \left(q_{y_{\max}}\left(e^{h_{n+1}} + \mu - e^{h_{y_{\max}}}\right) + (p_{n+1} - p_{y_{\max}} + q_{y_{\max}})\left(e^{h_{y_{\max}}} - \mu - e^{h_{n+1}}\right)\right)
$$

$$
= (p_{y_{\max}} - p_{n+1})\frac{e^{h_{n+1}}}{\sum_{y' \in \overline{y}} e^{h_{y'}}} \qquad (\mu = e^{h_{y_{\max}}} \text{ achieves the maximum})
$$

$$
\geq \frac{1}{n+1}(p_{y_{\max}} - p_{n+1}) \qquad (\text{by the assumption } h_{n+1} \geq h_{y_{\max}} = h_{h_{\max}})
$$

$$
= \frac{1}{n+1}(\Delta \mathcal{C}_{\mathsf{L_{def}}, \mathcal{H}}(h, x)) \qquad (\text{by the assumption } p_{y_{\max}} \geq p_{n+1} \text{ and } h_{y_{\max}} - h_{n+1} \leq 0)
$$

2. **Case II**: If $p_{y_{\max}} - p_{n+1} \le 0$ and $h_{y_{\max}} - h_{n+1} \ge 0$: we define a new hypothesis $h_\mu$ such that

$$h_\mu(x,y) = \begin{cases} \log\left(e^{h_{n+1}} - \mu\right) & y = y_{\max} \\ \log\left(e^{h_{y_{\max}}} + \mu\right) & y = n+1 \\ h(x,y) & \text{otherwise.} \end{cases}, \text{ where } e^{h_{n+1}} \ge \mu \ge 0. \text{ Then, we can lower bound}$$

the conditional regret of $\mathsf{L}_{\text{RL2D}}$ by using $\Delta \mathcal{C}_{\mathsf{L}_{\text{RL2D}},\mathcal{H}}(h,x) \ge \mathcal{C}_{\mathsf{L}_{\text{RL2D}}}(h) - \mathcal{C}^*_{\mathsf{L}_{\text{RL2D}}}(h_\mu)$ for any $e^{h_{n+1}} \ge \mu \ge 0$:

$$\Delta \mathcal{C}_{\mathsf{L}_{\text{RL2D}},\mathcal{H}}(h,x)$$

$$\ge \sup_{e^{h_{n+1}} \ge \mu \ge 0} \left( \mathcal{C}_{\mathsf{L}_{\text{RL2D}}}(h) - \mathcal{C}^*_{\mathsf{L}_{\text{RL2D}}}(h_\mu) \right)$$

$$\ge \sup_{e^{h_{n+1}} \ge \mu \ge 0} \left( q_{y_{\max}}\left(1 - \frac{e^{h_{y_{\max}}}}{\sum_{y' \in \overline{\mathcal{Y}}} e^{h_{y'}}}\right) + (p_{y_{\max}} - q_{y_{\max}})\left(1 - \frac{e^{h_{y_{\max}}} + e^{h_{n+1}}}{\sum_{y' \in \overline{\mathcal{Y}}} e^{h_{y'}}}\right)\right.$$

$$+ \sum_{y' \in \mathcal{Y}, y' \ne y_{\max}} (p_{y'} - q_{y'})\left(1 - \frac{e^{h_{y'}} + e^{h_{n+1}}}{\sum_{y' \in \overline{\mathcal{Y}}} e^{h_{y'}}}\right)$$

$$- q_{y_{\max}}\left(1 - \frac{e^{h_{n+1}} - \mu}{\sum_{y' \in \overline{\mathcal{Y}}} e^{h_{y'}}}\right) - (p_{y_{\max}} - q_{y_{\max}})\left(1 - \frac{e^{h_{n+1}} + e^{h_{h_{\max}}}}{\sum_{y' \in \overline{\mathcal{Y}}} e^{h_{y'}}}\right)\bigg)$$

$$\left. - \sum_{y' \in \mathcal{Y}, y' \ne y_{\max}} (p_{y'} - q_{y'})\left(1 - \frac{e^{h_{y'}} + e^{h_{y_{\max}}} + \mu}{\sum_{y' \in \overline{\mathcal{Y}}} e^{h_{y'}}}\right)\right)$$

$$= \frac{1}{\sum_{y' \in \overline{\mathcal{Y}}} e^{h_{y'}}} \sup_{e^{h_{n+1}} \ge \mu \ge 0} \left( q_{y_{\max}}\left(e^{h_{n+1}} - \mu - e^{h_{y_{\max}}}\right) + (p_{n+1} - p_{y_{\max}} + q_{y_{\max}})\left(e^{h_{y_{\max}}} + \mu - e^{h_{n+1}}\right)\right)$$

$$= (p_{n+1} - p_{y_{\max}}) \frac{e^{h_{y_{\max}}}}{\sum_{y' \in \overline{\mathcal{Y}}} e^{h_{y'}}} \qquad\qquad (\mu = e^{h_{n+1}} \text{ achieves the maximum})$$

$$\ge \frac{1}{n+1}(p_{n+1} - p_{y_{\max}}) \qquad\qquad\qquad (\text{by the assumption } h_{h_{\max}} = h_{y_{\max}} \ge h_{n+1})$$

$$= \frac{1}{n+1}(\Delta \mathcal{C}_{\mathsf{L}_{\text{def}},\mathcal{H}}(h,x)) \qquad (\text{by the assumption } p_{n+1} \ge p_{y_{\max}} \text{ and } h_{y_{\max}} - h_{n+1} \ge 0)$$

This proves the inequality (6). By Theorem B.1, we complete the proof. $\qquad\qquad\square$

**B.3** $\Psi(t) = -\log(t)$

**Theorem B.4.** *Assume that $\mathcal{H}$ is symmetric and complete. Assume that $c(x,y) = 1_{\mathsf{g}(x) \neq y}$. Then, for all $h \in \mathcal{H}$ and any distribution, the following $\mathcal{H}$-consistency bound holds:*

$$\mathcal{E}_{\mathsf{L}_{\mathrm{def}}}(h) - \mathcal{E}_{\mathsf{L}_{\mathrm{def}}}(\mathcal{H}) + \mathcal{M}_{\mathsf{L}_{\mathrm{def}}}(\mathcal{H}) \leq 2\sqrt{\mathcal{E}_{\mathsf{L}_{\mathrm{RL2D}}}(h) - \mathcal{E}_{\mathsf{L}_{\mathrm{RL2D}}}(\mathcal{H}) + \mathcal{M}_{\mathsf{L}_{\mathrm{RL2D}}}(\mathcal{H})}.$$

*Proof.* We can write the conditional error of the surrogate loss as follows:

$$
\begin{aligned}
&\mathcal{C}_{\mathsf{L}_{\mathrm{RL2D}}}(h,x) \\
&= \sum_{y \in \mathcal{Y}} p(x,y) \mathsf{L}_{\mathrm{RL2D}}(h,x,y) \\
&= -\sum_{y \in \mathcal{Y}} p(x,y) c(x,y) \log\left(\frac{e^{h(x,y)}}{\sum_{y' \in \overline{\mathcal{Y}}} e^{h(x,y')}}\right) - \sum_{y \in \mathcal{Y}} p(x,y)(1-c(x,y)) \log\left(\frac{e^{h(x,y)} + e^{h(x,n+1)}}{\sum_{y' \in \overline{\mathcal{Y}}} e^{h(x,y')}}\right) \\
&= -\sum_{y \in \mathcal{Y}} q_y \log\left(\frac{e^{h_y}}{\sum_{y' \in \overline{\mathcal{Y}}} e^{h_{y'}}}\right) - \sum_{y \in \mathcal{Y}} (p_y - q_y) \log\left(\frac{e^{h_y} + e^{h_{n+1}}}{\sum_{y' \in \overline{\mathcal{Y}}} e^{h_{y'}}}\right).
\end{aligned}
$$

By Lemma B.2, the conditional regret of the deferral loss can be expressed as

$$\Delta\mathcal{C}_{\mathsf{L}_{\mathrm{def}},\mathcal{H}}(h,x) = \max\{p_{y_{\max}}, p_{n+1}\} - p_{\mathsf{h}}.$$

Next, we will show that the conditional regret of the surrogate loss can be lower bounded as follows:

$$\Delta\mathcal{C}_{\mathsf{L}_{\mathrm{RL2D}},\mathcal{H}}(h,x) = \mathcal{C}_{\mathsf{L}_{\mathrm{RL2D}}}(h) - \mathcal{C}^*_{\mathsf{L}_{\mathrm{RL2D}}}(\mathcal{H}) \geq \frac{1}{2}(\Delta\mathcal{C}_{\mathsf{L}_{\mathrm{def}},\mathcal{H}}(h,x))^2. \tag{7}$$

We first consider the case where $\mathsf{g}(x) \neq y_{\max}$. Otherwise, it would be straightforward to see that the bound holds. In the case where $\mathsf{g}(x) \neq y_{\max}$, we have $q_{y_{\max}} = p_{y_{\max}}$. We first prove that for any hypothesis $h$ and $x \in \mathcal{X}$, if $y_{\max} \neq h_{\max}$, then the conditional error of $h$ can be lower bounded by that of $\overline{h}$, which satisfies that $\overline{h}(x,y) = \begin{cases} h_{h_{\max}} & y = y_{\max} \\ h_{y_{\max}} & y = h_{\max} \\ h_y & \text{otherwise.} \end{cases}$ . Indeed,

$$
\begin{aligned}
&\mathcal{C}_{\mathsf{L}_{\mathrm{RL2D}}}(h) - \mathcal{C}_{\mathsf{L}_{\mathrm{RL2D}}}(\overline{h}) \\
&= -q_{y_{\max}} \log\left(\frac{e^{h_{y_{\max}}}}{\sum_{y' \in \overline{\mathcal{Y}}} e^{h_{y'}}}\right) - (p_{y_{\max}} - q_{y_{\max}}) \log\left(\frac{e^{h_{y_{\max}}} + e^{h_{n+1}}}{\sum_{y' \in \overline{\mathcal{Y}}} e^{h_{y'}}}\right) \\
&\quad - q_{h_{\max}} \log\left(\frac{e^{h_{h_{\max}}}}{\sum_{y' \in \overline{\mathcal{Y}}} e^{h_{y'}}}\right) - (p_{h_{\max}} - q_{h_{\max}}) \log\left(\frac{e^{h_{h_{\max}}} + e^{h_{n+1}}}{\sum_{y' \in \overline{\mathcal{Y}}} e^{h_{y'}}}\right) \\
&\quad + q_{y_{\max}} \log\left(\frac{e^{h_{h_{\max}}}}{\sum_{y' \in \overline{\mathcal{Y}}} e^{h_{y'}}}\right) + (p_{y_{\max}} - q_{y_{\max}}) \log\left(\frac{e^{h_{h_{\max}}} + e^{h_{n+1}}}{\sum_{y' \in \overline{\mathcal{Y}}} e^{h_{y'}}}\right) \\
&\quad + q_{h_{\max}} \log\left(\frac{e^{h_{y_{\max}}}}{\sum_{y' \in \overline{\mathcal{Y}}} e^{h_{y'}}}\right) + (p_{h_{\max}} - q_{h_{\max}}) \log\left(\frac{e^{h_{y_{\max}}} + e^{h_{n+1}}}{\sum_{y' \in \overline{\mathcal{Y}}} e^{h_{y'}}}\right) \\
&= (q_{y_{\max}} - q_{h_{\max}}) \log\left(\frac{e^{h_{h_{\max}}}}{e^{h_{y_{\max}}}}\right) + (p_{y_{\max}} - q_{y_{\max}} - p_{h_{\max}} + q_{h_{\max}}) \log\left(\frac{e^{h_{h_{\max}}} + e^{h_{n+1}}}{e^{h_{y_{\max}}} + e^{h_{n+1}}}\right) \\
&\geq (p_{y_{\max}} - p_{h_{\max}}) \log\left(\frac{e^{h_{h_{\max}}} + e^{h_{n+1}}}{e^{h_{y_{\max}}} + e^{h_{n+1}}}\right) \\
&\geq 0.
\end{aligned}
$$

Therefore, we only need to lower bound the conditional regret of hypothesis $h$ satisfying $y_{\max} = h_{\max}$. Since $c(x,y) = 1_{\mathsf{g}(x) \neq y}$, we have $p_{y_{\max}} \geq p_{n+1} = p_{\mathsf{g}(x)}$. Note that when $(p_{y_{\max}} - p_{n+1})(h_{y_{\max}} - h_{n+1}) > 0$, we have $\Delta\mathcal{C}_{\mathsf{L}_{\mathrm{def}},\mathcal{H}}(h,x) = \max\{p_{y_{\max}}, p_{n+1}\} - p_{\mathsf{h}} = 0$. When $h_{y_{\max}} - h_{n+1} \leq 0$, we define a new hypothesis $h_\mu$ such that $h_\mu(x,y) = \begin{cases} \log(e^{h_{n+1}} + \mu) & y = y_{\max} \\ \log(e^{h_{y_{\max}}} - \mu) & y = n+1 \\ h(x,y) & \text{otherwise.} \end{cases}$ , where $e^{h_{y_{\max}}} - e^{h_{n+1}} \leq$

$\mu \le e^{h_{y_{\max}}}$. Then, we can lower bound the conditional regret of $\mathsf{L}_{\mathrm{RL2D}}$ by using $\Delta\mathcal{C}_{\mathsf{L}_{\mathrm{RL2D}},\mathcal{H}}(h,x) \ge \mathcal{C}_{\mathsf{L}_{\mathrm{RL2D}}}(h) - \mathcal{C}^*_{\mathsf{L}_{\mathrm{RL2D}}}(h_\mu)$ for any $e^{h_{y_{\max}}} - e^{h_{n+1}} \le \mu \le e^{h_{y_{\max}}}$:

$$\Delta\mathcal{C}_{\mathsf{L}_{\mathrm{RL2D}},\mathcal{H}}(h,x)$$
$$\ge \sup_{e^{h_{y_{\max}}} \ge \mu \ge e^{h_{y_{\max}}} - e^{h_{n+1}}} \left(\mathcal{C}_{\mathsf{L}_{\mathrm{RL2D}}}(h) - \mathcal{C}^*_{\mathsf{L}_{\mathrm{RL2D}}}(h_\mu)\right)$$
$$\ge \sup_{e^{h_{y_{\max}}} \ge \mu \ge e^{h_{y_{\max}}} - e^{h_{n+1}}} \left(-q_{y_{\max}} \log\left(\frac{e^{h_{y_{\max}}}}{\sum_{y' \in \overline{y}} e^{h_{y'}}}\right) - (p_{y_{\max}} - q_{y_{\max}}) \log\left(\frac{e^{h_{y_{\max}}} + e^{h_{n+1}}}{\sum_{y' \in \overline{y}} e^{h_{y'}}}\right)\right.$$
$$- \sum_{y' \in \mathcal{Y}, y' \ne y_{\max}} (p_{y'} - q_{y'}) \log\left(\frac{e^{h_{y'}} + e^{h_{n+1}}}{\sum_{y' \in \overline{y}} e^{h_{y'}}}\right)$$
$$+ q_{y_{\max}} \log\left(\frac{e^{h_{n+1}} + \mu}{\sum_{y' \in \overline{y}} e^{h_{y'}}}\right) + (p_{y_{\max}} - q_{y_{\max}}) \log\left(\frac{e^{h_{n+1}} + e^{h_{y_{\max}}}}{\sum_{y' \in \overline{y}} e^{h_{y'}}}\right)\right)$$
$$\left. + \sum_{y' \in \mathcal{Y}, y' \ne y_{\max}} (p_{y'} - q_{y'}) \log\left(\frac{e^{h_{y'}} + e^{h_{y_{\max}}} - \mu}{\sum_{y' \in \overline{y}} e^{h_{y'}}}\right)\right)$$
$$= \sup_{e^{h_{y_{\max}}} \ge \mu \ge e^{h_{y_{\max}}} - e^{h_{n+1}}} \left(q_{y_{\max}} \log\frac{e^{h_{n+1}} + \mu}{e^{h_{y_{\max}}}} + \sum_{y' \in \mathcal{Y}, y' \ne y_{\max}} (p_{y'} - q_{y'}) \log\frac{e^{h_{y'}} + e^{h_{y_{\max}}} - \mu}{e^{h_{y'}} + e^{h_{n+1}}}\right)$$
$$\ge \sup_{e^{h_{y_{\max}}} \ge \mu \ge e^{h_{y_{\max}}} - e^{h_{n+1}}} \left(q_{y_{\max}} \log\frac{e^{h_{n+1}} + \mu}{e^{h_{y_{\max}}}} + \sum_{y' \in \mathcal{Y}, y' \ne y_{\max}} (p_{y'} - q_{y'}) \log\frac{e^{h_{y_{\max}}} - \mu}{e^{h_{n+1}}}\right)$$
$$(e^{h_{y_{\max}}} - e^{h_{n+1}} \le \mu \le e^{h_{y_{\max}}})$$
$$= \sup_{e^{h_{y_{\max}}} \ge \mu \ge e^{h_{y_{\max}}} - e^{h_{n+1}}} \left(q_{y_{\max}} \log\frac{e^{h_{n+1}} + \mu}{e^{h_{y_{\max}}}} + (p_{n+1} - (p_{y_{\max}} - q_{y_{\max}})) \log\frac{e^{h_{y_{\max}}} - \mu}{e^{h_{n+1}}}\right).$$

By differentiating with respect to $\mu$, we obtain that

$$\mu = \frac{q_{y_{\max}} e^{h_{y_{\max}}} - (p_{n+1} - (p_{y_{\max}} - q_{y_{\max}})) e^{h_{n+1}}}{q_{y_{\max}} + (p_{n+1} - (p_{y_{\max}} - q_{y_{\max}}))}$$

achieves the maximum. Plugging it into the expression, we have

$$\Delta\mathcal{C}_{\mathsf{L}_{\mathrm{RL2D}},\mathcal{H}}(h,x)$$
$$\ge q_{y_{\max}} \log\left[\frac{[e^{h_{y_{\max}}} + e^{h_{n+1}}] q_{y_{\max}}}{e^{h_{y_{\max}}} [q_{y_{\max}} + (p_{n+1} - (p_{y_{\max}} - q_{y_{\max}}))]}\right]$$
$$+ (p_{n+1} - (p_{y_{\max}} - q_{y_{\max}})) \log\left[\frac{[e^{h_{y_{\max}}} + e^{h_{n+1}}](p_{n+1} - (p_{y_{\max}} - q_{y_{\max}}))}{e^{h_{n+1}} [q_{y_{\max}} + (p_{n+1} - (p_{y_{\max}} - q_{y_{\max}}))]}\right].$$

This can be further lower bounded by taking the minimum over $h \in \mathcal{H}$, where the minimum is attained when $e^{h_{y_{\max}}} = e^{h_{n+1}}$ Therefore,

$$\Delta\mathcal{C}_{\mathsf{L}_{\mathrm{RL2D}},\mathcal{H}}(h,x)$$
$$\ge q_{y_{\max}} \log\left[\frac{2 q_{y_{\max}}}{q_{y_{\max}} + (p_{n+1} - (p_{y_{\max}} - q_{y_{\max}}))}\right]$$
$$+ (p_{n+1} - (p_{y_{\max}} - q_{y_{\max}})) \log\left[\frac{2(p_{n+1} - (p_{y_{\max}} - q_{y_{\max}}))}{q_{y_{\max}} + (p_{n+1} - (p_{y_{\max}} - q_{y_{\max}}))}\right].$$

By applying Pinsker's inequality [Mohri et al., 2018, Proposition E.7], we obtain

$$\Delta \mathcal{C}_{\mathsf{L}_{\mathrm{RL2D}}, \mathcal{H}}(h, x)$$

$$\geq \left[ q_{y_{\max}} + p_{n+1} - (p_{y_{\max}} - q_{y_{\max}}) \right]$$

$$\times \frac{1}{2} \left[ \left| \frac{q_{y_{\max}}}{q_{y_{\max}} + p_{n+1} - (p_{y_{\max}} - q_{y_{\max}})} - \frac{1}{2} \right| + \left| \frac{p_{n+1} - (p_{y_{\max}} - q_{y_{\max}})}{q_{y_{\max}} + p_{n+1} - (p_{y_{\max}} - q_{y_{\max}})} - \frac{1}{2} \right| \right]^2$$

$$\geq \frac{1}{2} \frac{(p_{y_{\max}} - p_{n+1})^2}{q_{y_{\max}} + p_{n+1} - (p_{y_{\max}} - q_{y_{\max}})}$$

$$\geq \frac{1}{2} (p_{y_{\max}} - p_{n+1})^2 \qquad\qquad (q_{y_{\max}} + p_{n+1} - (p_{y_{\max}} - q_{y_{\max}}) \leq 1)$$

$$= \frac{1}{2} (\Delta \mathcal{C}_{\mathsf{L}_{\mathrm{def}}, \mathcal{H}}(h, x))^2 \qquad\qquad \text{(by the assumption } p_{y_{\max}} \geq p_{n+1} \text{ and } h_{y_{\max}} \leq h_{n+1})$$

This proves the inequality (7). In the case where $\mathsf{g}(x) = y_{\max}$, we have $p_{n+1} = p_{y_{\max}}$. By Lemma B.2, the conditional regret of the deferral loss can be expressed as $\Delta \mathcal{C}_{\mathsf{L}_{\mathrm{def}}, \mathcal{H}}(h, x) = p_{n+1} - p_{\mathsf{h}}$. If $\mathsf{h}(x) = n+1$, then we have $\Delta \mathcal{C}_{\mathsf{L}_{\mathrm{def}}, \mathcal{H}}(h, x) = 0$. Otherwise, when $\mathsf{h}(x) \neq n+1$, we can proceed in the similar way as above, by defining a new hypothesis $h_\mu$ such that $h_\mu(x, y) = \begin{cases} \log(e^{h_{n+1}} + \mu) & y = \mathsf{h}(x) \\ \log(e^{h_{\mathsf{h}(x)}} - \mu) & y = n+1 \\ h(x, y) & \text{otherwise} \end{cases}$.

Then, we can lower bound the conditional regret of $\mathsf{L}_{\mathrm{RL2D}}$ by using $\Delta \mathcal{C}_{\mathsf{L}_{\mathrm{RL2D}}, \mathcal{H}}(h, x) \geq \mathcal{C}_{\mathsf{L}_{\mathrm{RL2D}}}(h) - \mathcal{C}^*_{\mathsf{L}_{\mathrm{RL2D}}}(h_\mu)$, by applying the same derivation as above, modulo replacing $y_{\max}$ with $\mathsf{h}(x)$. This leads to the inequality (7) as well. By Theorem B.1, we complete the proof. $\qquad \square$

**B.4**   $\Psi(t) = \frac{1}{q}(1 - t^q)$

**Theorem B.5.** *Assume that $\mathcal{H}$ is symmetric and complete. Assume that $c(x,y) = 1_{g(x)\neq y}$. Then, for all $h \in \mathcal{H}$ and any distribution, the following $\mathcal{H}$-consistency bound holds:*

$$\mathcal{E}_{\mathsf{L}_{\mathrm{def}}}(h) - \mathcal{E}_{\mathsf{L}_{\mathrm{def}}}(\mathcal{H}) + \mathcal{M}_{\mathsf{L}_{\mathrm{def}}}(\mathcal{H}) \leq 2\sqrt{(n+1)^\alpha(\mathcal{E}_{\mathsf{L}_{\mathrm{RL2D}}}(h) - \mathcal{E}_{\mathsf{L}_{\mathrm{RL2D}}}(\mathcal{H}) + \mathcal{M}_{\mathsf{L}_{\mathrm{RL2D}}}(\mathcal{H}))}.$$

*Proof.* We can write the conditional error of the surrogate loss as follows:

$$\mathcal{C}_{\mathsf{L}_{\mathrm{RL2D}}}(h,x) = \sum_{y \in \mathcal{Y}} p(x,y)\mathsf{L}_{\mathrm{RL2D}}(h,x,y)$$

$$= \frac{1}{q}\sum_{y \in \mathcal{Y}} p(x,y)c(x,y)\left(1 - \left(\frac{e^{h(x,y)}}{\sum_{y' \in \overline{y}} e^{h(x,y')}}\right)^q\right)$$

$$+ \frac{1}{q}\sum_{y \in \mathcal{Y}} p(x,y)(1 - c(x,y))\left(1 - \left(\frac{e^{h(x,y)} + e^{h(x,n+1)}}{\sum_{y' \in \overline{y}} e^{h(x,y')}}\right)^q\right)$$

$$= \frac{1}{q}\sum_{y \in \mathcal{Y}} q_y\left(1 - \left(\frac{e^{h_y}}{\sum_{y' \in \overline{y}} e^{h_{y'}}}\right)^q\right) + \frac{1}{q}\sum_{y \in \mathcal{Y}}(p_y - q_y)\left(1 - \left(\frac{e^{h_y} + e^{h_{n+1}}}{\sum_{y' \in \overline{y}} e^{h_{y'}}}\right)^q\right).$$

By Lemma B.2, the conditional regret of the deferral loss can be expressed as

$$\Delta\mathcal{C}_{\mathsf{L}_{\mathrm{def}},\mathcal{H}}(h,x) = \max\{p_{y_{\max}}, p_{n+1}\} - p_{\mathsf{h}}.$$

Next, we will show that the conditional regret of the surrogate loss can be lower bounded as follows:

$$\Delta\mathcal{C}_{\mathsf{L}_{\mathrm{RL2D}},\mathcal{H}}(h,x) = \mathcal{C}_{\mathsf{L}_{\mathrm{RL2D}}}(h) - \mathcal{C}^*_{\mathsf{L}_{\mathrm{RL2D}}}(\mathcal{H}) \geq \frac{1}{2(n+1)^q}(\Delta\mathcal{C}_{\mathsf{L}_{\mathrm{def}},\mathcal{H}}(h,x))^2. \qquad (8)$$

We first consider the case where $g(x) \neq y_{\max}$. Otherwise, it would be straightforward to see that the bound holds. In the case where $g(x) \neq y_{\max}$, we have $q_{y_{\max}} = p_{y_{\max}}$. We first prove that for any hypothesis $h$ and $x \in \mathcal{X}$, if $y_{\max} \neq h_{\max}$, then the conditional error of $h$ can be lower bounded by that of $\overline{h}$, which satisfies that $\overline{h}(x,y) = \begin{cases} h_{h_{\max}} & y = y_{\max} \\ h_{y_{\max}} & y = h_{\max} \\ h_y & \text{otherwise.} \end{cases}$ . Indeed,

$$q\left(\mathcal{C}_{\mathsf{L}_{\mathrm{RL2D}}}(h) - \mathcal{C}_{\mathsf{L}_{\mathrm{RL2D}}}(\overline{h})\right)$$

$$= q_{y_{\max}}\left(1 - \left(\frac{e^{h_{y_{\max}}}}{\sum_{y' \in \overline{y}} e^{h_{y'}}}\right)^q\right) + (p_{y_{\max}} - q_{y_{\max}})\left(1 - \left(\frac{e^{h_{y_{\max}}} + e^{h_{n+1}}}{\sum_{y' \in \overline{y}} e^{h_{y'}}}\right)^q\right)$$

$$+ q_{h_{\max}}\left(1 - \left(\frac{e^{h_{h_{\max}}}}{\sum_{y' \in \overline{y}} e^{h_{y'}}}\right)^q\right) + (p_{h_{\max}} - q_{h_{\max}})\left(1 - \left(\frac{e^{h_{h_{\max}}} + e^{h_{n+1}}}{\sum_{y' \in \overline{y}} e^{h_{y'}}}\right)^q\right)$$

$$- q_{y_{\max}}\left(1 - \left(\frac{e^{h_{h_{\max}}}}{\sum_{y' \in \overline{y}} e^{h_{y'}}}\right)^q\right) - (p_{y_{\max}} - q_{y_{\max}})\left(1 - \left(\frac{e^{h_{h_{\max}}} + e^{h_{n+1}}}{\sum_{y' \in \overline{y}} e^{h_{y'}}}\right)^q\right)$$

$$- q_{h_{\max}}\left(1 - \left(\frac{e^{h_{y_{\max}}}}{\sum_{y' \in \overline{y}} e^{h_{y'}}}\right)^q\right) + (p_{h_{\max}} - q_{h_{\max}})\left(1 - \left(\frac{e^{h_{y_{\max}}} + e^{h_{n+1}}}{\sum_{y' \in \overline{y}} e^{h_{y'}}}\right)^q\right)$$

$$= (q_{y_{\max}} - q_{h_{\max}})\left[\left(1 - \left(\frac{e^{h_{y_{\max}}}}{\sum_{y' \in \overline{y}} e^{h_{y'}}}\right)^q\right) - \left(1 - \left(\frac{e^{h_{h_{\max}}}}{\sum_{y' \in \overline{y}} e^{h_{y'}}}\right)^q\right)\right]$$

$$+ (p_{y_{\max}} - q_{y_{\max}} - p_{h_{\max}} + q_{h_{\max}})\left[\left(1 - \left(\frac{e^{h_{y_{\max}}} + e^{h_{n+1}}}{\sum_{y' \in \overline{y}} e^{h_{y'}}}\right)^q\right) - \left(1 - \left(\frac{e^{h_{h_{\max}}} + e^{h_{n+1}}}{\sum_{y' \in \overline{y}} e^{h_{y'}}}\right)^q\right)\right]$$

$$\geq (p_{y_{\max}} - p_{h_{\max}})\left[\left(1 - \left(\frac{e^{h_{y_{\max}}} + e^{h_{n+1}}}{\sum_{y' \in \overline{y}} e^{h_{y'}}}\right)^q\right) - \left(1 - \left(\frac{e^{h_{h_{\max}}} + e^{h_{n+1}}}{\sum_{y' \in \overline{y}} e^{h_{y'}}}\right)^q\right)\right]$$

$$\geq 0.$$

Therefore, we only need to lower bound the conditional regret of hypothesis $h$ satisfying $y_{\max} = h_{\max}$. Since $c(x, y) = 1_{g(x) \neq y}$, we have $p_{y_{\max}} \geq p_{n+1} = p_{g(x)}$. Note that when $(p_{y_{\max}} - p_{n+1})(h_{y_{\max}} - h_{n+1}) > 0$, we have $\Delta \mathcal{C}_{\mathsf{L}_{\mathrm{def}}, \mathcal{H}}(h, x) = \max\{p_{y_{\max}}, p_{n+1}\} - p_h = 0$. When $h_{y_{\max}} - h_{n+1} \leq 0$, we define a new hypothesis $h_\mu$ such that $h_\mu(x, y) = \begin{cases} \log(e^{h_{n+1}} + \mu) & y = y_{\max} \\ \log(e^{h_{y_{\max}}} - \mu) & y = n+1 \\ h(x, y) & \text{otherwise.} \end{cases}$, where $e^{h_{y_{\max}}} - e^{h_{n+1}} \leq \mu \leq e^{h_{y_{\max}}}$. Then, we can lower bound the conditional regret of hypothesis $h$ by using $\Delta \mathcal{C}_{\mathsf{L}_{\mathrm{RL2D}}, \mathcal{H}}(h, x) \geq \mathcal{C}_{\mathsf{L}_{\mathrm{RL2D}}}(h) - \mathcal{C}^*_{\mathsf{L}_{\mathrm{RL2D}}}(h_\mu)$ for any $e^{h_{y_{\max}}} - e^{h_{n+1}} \leq \mu \leq e^{h_{y_{\max}}}$:

$$\Delta \mathcal{C}_{\mathsf{L}_{\mathrm{RL2D}}, \mathcal{H}}(h, x)$$

$$\geq \sup_{e^{h_{y_{\max}}} \geq \mu \geq e^{h_{y_{\max}}} - e^{h_{n+1}}} \left( \mathcal{C}_{\mathsf{L}_{\mathrm{RL2D}}}(h) - \mathcal{C}^*_{\mathsf{L}_{\mathrm{RL2D}}}(h_\mu) \right)$$

$$\geq \frac{1}{q} \sup_{e^{h_{y_{\max}}} \geq \mu \geq e^{h_{y_{\max}}} - e^{h_{n+1}}} \left( q_{y_{\max}} \left( 1 - \left( \frac{e^{h_{y_{\max}}}}{\sum_{y' \in \overline{y}} e^{h_{y'}}} \right)^q \right) + (p_{y_{\max}} - q_{y_{\max}}) \left( 1 - \left( \frac{e^{h_{y_{\max}}} + e^{h_{n+1}}}{\sum_{y' \in \overline{y}} e^{h_{y'}}} \right)^q \right) \right.$$

$$+ \sum_{y' \in \mathcal{Y}, y' \neq y_{\max}} (p_{y'} - q_{y'}) \left( 1 - \left( \frac{e^{h_{y'}} + e^{h_{n+1}}}{\sum_{y' \in \overline{y}} e^{h_{y'}}} \right)^q \right)$$

$$- q_{y_{\max}} \left( 1 - \left( \frac{e^{h_{n+1}} + \mu}{\sum_{y' \in \overline{y}} e^{h_{y'}}} \right)^q \right) - (p_{y_{\max}} - q_{y_{\max}}) \left( 1 - \left( \frac{e^{h_{n+1}} + e^{h_{y_{\max}}}}{\sum_{y' \in \overline{y}} e^{h_{y'}}} \right) \right)^q$$

$$\left. - \sum_{y' \in \mathcal{Y}, y' \neq y_{\max}} (p_{y'} - q_{y'}) \left( 1 - \left( \frac{e^{h_{y'}} + e^{h_{y_{\max}}} - \mu}{\sum_{y' \in \overline{y}} e^{h_{y'}}} \right)^q \right) \right)$$

$$\geq \frac{1}{q} \sup_{e^{h_{y_{\max}}} \geq \mu \geq e^{h_{y_{\max}}} - e^{h_{n+1}}} \left( q_{y_{\max}} \left( 1 - \left( \frac{e^{h_{y_{\max}}}}{\sum_{y' \in \overline{y}} e^{h_{y'}}} \right)^q \right) + \sum_{y' \in \mathcal{Y}, y' \neq y_{\max}} (p_{y'} - q_{y'}) \left( 1 - \left( \frac{e^{h_{n+1}}}{\sum_{y' \in \overline{y}} e^{h_{y'}}} \right)^q \right) \right.$$

$$\left. - q_{y_{\max}} \left( 1 - \left( \frac{e^{h_{n+1}} + \mu}{\sum_{y' \in \overline{y}} e^{h_{y'}}} \right)^q \right) - \sum_{y' \in \mathcal{Y}, y' \neq y_{\max}} (p_{y'} - q_{y'}) \left( 1 - \left( \frac{e^{h_{y_{\max}}} - \mu}{\sum_{y' \in \overline{y}} e^{h_{y'}}} \right)^q \right) \right)$$
$$(e^{h_{y_{\max}}} - e^{h_{n+1}} \leq \mu \leq e^{h_{y_{\max}}})$$

$$= \frac{1}{q} \sup_{e^{h_{y_{\max}}} \geq \mu \geq e^{h_{y_{\max}}} - e^{h_{n+1}}} \left( q_{y_{\max}} \left( 1 - \left( \frac{e^{h_{y_{\max}}}}{\sum_{y' \in \overline{y}} e^{h_{y'}}} \right)^q \right) + (p_{n+1} - (p_{y_{\max}} - q_{y_{\max}})) \left( 1 - \left( \frac{e^{h_{n+1}}}{\sum_{y' \in \overline{y}} e^{h_{y'}}} \right)^q \right) \right.$$

$$\left. - q_{y_{\max}} \left( 1 - \left( \frac{e^{h_{n+1}} + \mu}{\sum_{y' \in \overline{y}} e^{h_{y'}}} \right)^q \right) - (p_{n+1} - (p_{y_{\max}} - q_{y_{\max}})) \left( 1 - \left( \frac{e^{h_{y_{\max}}} - \mu}{\sum_{y' \in \overline{y}} e^{h_{y'}}} \right)^q \right) \right)$$

By differentiating with respect to $\mu$, we obtain that

$$\mu = \frac{(p_{n+1} - (p_{y_{\max}} - q_{y_{\max}}))^{\frac{1}{q-1}} e^{h_{y_{\max}}} - (q_{y_{\max}})^{\frac{1}{q-1}} e^{h_{n+1}}}{(q_{y_{\max}})^{\frac{1}{q-1}} + (p_{n+1} - (p_{y_{\max}} - q_{y_{\max}}))^{\frac{1}{q-1}}}$$

achieves the maximum. Plugging it into the expression, we have

$$\Delta \mathcal{C}_{\mathsf{L}_{\mathrm{RL2D}}, \mathcal{H}}(h, x)$$

$$\geq \frac{1}{q} \left( -q_{y_{\max}} \left( \frac{e^{h_{y_{\max}}}}{\sum_{y' \in \overline{y}} e^{h_{y'}}} \right)^q - (p_{n+1} - (p_{y_{\max}} - q_{y_{\max}})) \left( \frac{e^{h_{n+1}}}{\sum_{y' \in \overline{y}} e^{h_{y'}}} \right)^q \right.$$

$$+ q_{y_{\max}} \left[ \frac{[e^{h_{y_{\max}}} + e^{h_{n+1}}](p_{n+1} - (p_{y_{\max}} - q_{y_{\max}}))^{\frac{1}{q-1}}}{\sum_{y' \in \overline{y}} e^{h_{y'}} \left[ q_{y_{\max}}^{\frac{1}{q-1}} + (p_{n+1} - (p_{y_{\max}} - q_{y_{\max}}))^{\frac{1}{q-1}} \right]} \right]^q$$

$$\left. + (p_{n+1} - (p_{y_{\max}} - q_{y_{\max}})) \left[ \frac{[e^{h_{y_{\max}}} + e^{h_{n+1}}] q_{y_{\max}}^{\frac{1}{q-1}}}{\sum_{y' \in \overline{y}} e^{h_{y'}} \left[ q_{y_{\max}}^{\frac{1}{q-1}} + (p_{n+1} - (p_{y_{\max}} - q_{y_{\max}}))^{\frac{1}{q-1}} \right]} \right]^q \right)$$

This can be further lower bounded by taking the minimum over $h \in \mathcal{H}$, where the minimum is attained when $e^{h_{n+1}} = e^{h_{y_{\max}}} = e^{h_y}$ for all $y \in \mathcal{Y}$. Therefore,

$$\Delta \mathcal{C}_{\mathsf{L}_{\mathrm{RL2D}}, \mathcal{H}}(h, x) \geq \frac{2}{(n+1)^q} \left( \left[ \frac{q_{y_{\max}}^{\frac{1}{1-q}} + (p_{n+1} - (p_{y_{\max}} - q_{y_{\max}}))^{\frac{1}{1-q}}}{2} \right]^{1-q} - \frac{p_{n+1} - p_{y_{\max}}}{2} \right)$$

$$\text{(minimum is attained when } e^{h_{n+1}} = e^{h_{y_{\max}}} = e^{h_y}, \forall y \in \mathcal{Y})$$

$$\geq \frac{1}{2(n+1)^q} (p_{y_{\max}} - p_{n+1})^2$$

$$(q_{y_{\max}} + (p_{n+1} - (p_{y_{\max}} - q_{y_{\max}}))) \leq 1 \text{ and by analyzing the Taylor expansion)}$$

$$= \frac{1}{2(n+1)^q} (\Delta \mathcal{C}_{\mathsf{L}_{\mathrm{def}}, \mathcal{H}}(h, x))^2 \qquad (p_{y_{\max}} \geq p_{n+1} \text{ and } h_{y_{\max}} \leq h_{n+1})$$

This proves the inequality (8). In the case where $\mathsf{g}(x) = y_{\max}$, we have $p_{n+1} = p_{y_{\max}}$. By Lemma B.2, the conditional regret of the deferral loss can be expressed as $\Delta \mathcal{C}_{\mathsf{L}_{\mathrm{def}}, \mathcal{H}}(h, x) = p_{n+1} - p_{\mathsf{h}}$. If $\mathsf{h}(x) = n+1$, then we have $\Delta \mathcal{C}_{\mathsf{L}_{\mathrm{def}}, \mathcal{H}}(h, x) = 0$. Otherwise, when $\mathsf{h}(x) \neq n+1$, we can proceed in the similar way as above, by defining a new hypothesis $h_\mu$ such that $h_\mu(x, y) = \begin{cases} \log(e^{h_{n+1}} + \mu) & y = \mathsf{h}(x) \\ \log(e^{h_{\mathsf{h}(x)}} - \mu) & y = n+1 \\ h(x, y) & \text{otherwise} \end{cases}$.

Then, we can lower bound the conditional regret of $\mathsf{L}_{\mathrm{RL2D}}$ by using $\Delta \mathcal{C}_{\mathsf{L}_{\mathrm{RL2D}}, \mathcal{H}}(h, x) \geq \mathcal{C}_{\mathsf{L}_{\mathrm{RL2D}}}(h) - \mathcal{C}^*_{\mathsf{L}_{\mathrm{RL2D}}}(h_\mu)$, by applying the same derivation as above, modulo replacing $y_{\max}$ with $\mathsf{h}(x)$. This leads to the inequality (8) as well. By Theorem B.1, we complete the proof. $\qquad \square$

## C  Proof of Theorem 5.1

**Theorem 5.1.** *Assume that there exists a zero error solution $h^* \in \mathcal{H}$ with $\mathcal{E}_{\ell_{0-1}}(h^*) = 0$ and $\mathcal{H}$ is closed under scaling. Assume that $\lim_{t \to 1} \Psi(t) = 0$. Then, the minimizability gap of comp-sum loss $\ell_{\mathrm{comp}}$ vanishes: $\mathcal{M}_{\ell_{\mathrm{comp}}}(\mathcal{H}) = 0$.*

*Proof.* By definition and the Lebesgue dominated convergence theorem, we have

$$\mathcal{M}_{\ell_{\mathrm{comp}}}(\mathcal{H}) \leq \mathcal{E}^*_{\ell_{\mathrm{comp}}}(\mathcal{H}) \leq \lim_{\alpha \to +\infty} \mathbb{E}\left[ \Psi\left( \frac{e^{\alpha h^*(x,y)}}{\sum_{y' \in \mathcal{Y}} e^{\alpha h^*(x,y')}} \right) \right] = 0.$$

This completes the proof. $\qquad \square$

## D  Future work

While we presented a comprehensive study of surrogate loss functions for learning to defer, our work focused on the standard single-expert and single-stage setting, aligning with previous work [Mozannar et al., 2023]. However, an interesting direction is to extend our approach to multi-expert [Verma et al., 2023] and two-stage settings [Mao et al., 2023a], which we have left for future work.

