# OpenReview forum: "Realizable $H$-Consistent and Bayes-Consistent Loss Functions for Learning to Defer"
_NeurIPS.cc/2024/Conference — NeurIPS 2024 poster_

### Official Review · Reviewer_W4md · 2024-07-09

**Soundness:** 3
**Presentation:** 3
**Contribution:** 3
**Rating:** 7
**Confidence:** 3

**Summary:**

This work studies learning to defer, focusing on the single-stage and single-expert setting. It introduces a family of surrogate losses based on comp-sum losses [Mao et al., 2023b] and establishes their realizable H-consistency (under mild conditions). In addition, when the base loss is the logistic loss $\Psi_{log}$ or the generalized cross entropy loss $\Psi_{gce}$, H-consistency bounds are proven (under mild conditions) when the cost function of deferring is classification error. When the base loss is the mean absolute error loss $\Psi_{mae}$, H-consistency bounds are proven (under mild conditions) when the cost function of deferring is general. Note that H-consistency implies Bayes consistency. The results also close an open question raised by Mozannar et al., 2023. The relationship between realizable H-consistency and H-consistency bounds is then analyzed for learning to defer. Finally, the proposed surrogates are evaluated empirically and compared with existing baselines.

**Strengths:**

**Originality**

- A new family of surrogate losses based on comp-sum losses (focusing on $\Psi_{log}$, $\Psi_{gce}$, and $\Psi_{mae}$) [Mao et al., 2023b] for learning to defer (the single-stage and single-expert setting).
- Conditions for realizable H-consistency are identified (Theorem 4.1). In particular, $\Psi_{log}$, $\Psi_{gce}$, and $\Psi_{mae}$ all satisfy the conditions. This result is more general than Mozannar et al. [2023]
- When the cost function of deferring is classification error, H-consistency bounds are proven for base losses $\Psi_{log}$ and $\Psi_{gce}$ (Theorem 4.2).
- When the cost function of deferring is general, an H-consistency bound is proven for base loss $\Psi_{mae}$ (Theorem 4.3).

The results also close an open question raised by Mozannar et al., 2023 (Section 4.4). The relationship between realizable H-consistency and H-consistency bounds is then analyzed for learning to defer (Section 5). Related work is adequately cited and compared.

**Quality**

The submission is technically sound. Claims are well supported by proofs.

**Clarity**

The submission is generally clearly written and well organized.

**Significance**

The work closes an open question raised in previous work. Its work to connect both realizable H-consistency and H-consistency bounds might be useful in other learning settings.

**Weaknesses:**

**Originality**

- The realizable H-consistency result (Theorem 4.1) only applies to a subset of comp-sum losses.
- The H-consistency results (Theorems 4.2 and 4.3) require case-by-case analysis. Is it possible to prove such results for all comp-sum losses?

**Quality**

- Some experiments in realizable settings can help confirm the realizable H-consistency result.

**Clarity**

- Section 5 is a bit unclear. Lines 300-302: Do you mean that in the realizable setting, all surrogate minimizability gaps vanish? Or is it only for comp-sum losses?

**Significance**

- The applicability of this work might be limited to learning to defer.

**Questions:**

Besides my concerns above, here are some other questions:

1. Can constrained losses [1] be used as the base loss?

2. Lines 273-274: Can negative results be proven formally?


[1] Y. Lee, Y. Lin, and G. Wahba. Multicategory support vector machines: Theory and application to the classification of microarray data and satellite radiance data. Journal of the American Statistical Association, 99(465):67–81, 2004.

**Limitations:**

The authors have adequately addressed the limitations.

---

> ### Author Rebuttal · Authors · 2024-08-06
>
> Thank you for your appreciation of our work. We will take your suggestions into account when preparing the final version. Below please find responses to specific questions.
>
> **Weaknesses:**
>
> **1. The realizable H-consistency result (Theorem 4.1) only applies to a subset of comp-sum losses.**
>
> **Response:** We would like to clarify that the assumption regarding the function $\Phi$ in Theorem 4.1 is mild. The condition that $\Psi$ is non-increasing, with $\Psi(\frac{2}{3}) > 0$ and $\lim_{t \to 1} \Psi(t) = 0$, is satisfied by all common comp-sum losses in practice.
>
> **2. The H-consistency results (Theorems 4.2 and 4.3) require case-by-case analysis. Is it possible to prove such results for all comp-sum losses?**
>
> **Response:** That's a great question. We expect that our proof ideas can potentially be extended to other comp-sum losses. First, we demonstrate that for any hypothesis $h$ and input $x$, if $y_{\max}$, the label with the highest conditional probability, is not the predicted label $h_{\max}$, the conditional error of $h$ is lower bounded by a modified hypothesis $\overline{h}$ (obtained by swapping the scores of $y_{\max}$ and $h_{\max}$). Then, we show that for hypotheses where $y_{\max} = h_{\max}$, we lower bound their conditional regret in terms of the conditional regret of the deferral loss using a new hypothesis $h_{\mu}$.
>
> However, the proof of establishing lower bounds in each case depends on the forms of the comp-sum losses and indeed requires a case-by-case analysis. Presenting a unified analysis and extending these results to all comp-sum losses is left as future work. Nevertheless, our results have included the most widely used comp-sum losses in practice, such as logistic loss, generalized cross-entropy loss, and mean absolute error loss.
>
> **3. Some experiments in realizable settings can help confirm the realizable H-consistency result.**
>
> **Response:** Thank you for the feedback. We have included the additional experiment in the realizable case. The additional experimental result (Figure 1 in the global response) shows that our surrogate loss $\mathsf L_{\mathrm{RL2D}}$ with $q = 0.7$ and $q = 1$ are realizable $H$-consistent, while $\mathsf L_{\mathrm{CE}}$, $\mathsf L_{\mathrm{OVA}}$ and $\mathsf L_{\mathrm{general}}$ are not. This validates our theory.
>
> **4. Section 5 is a bit unclear. Lines 300-302: Do you mean that in the realizable setting, all surrogate minimizability gaps vanish? Or is it only for comp-sum losses?**
>
> **Response:** In standard classification, minimizability gaps vanish under the realizable assumption for common multi-class surrogate losses such as max losses [Crammer and Singer, 2001], sum losses [Weston and Watkins, 1999], and comp-sum losses under mild assumptions. We will further clarify this in the final version.
>
> [1] K. Crammer and Y. Singer. On the algorithmic implementation of multiclass kernel-based vector machines. Journal of machine learning research, 2(Dec):265–292, 2001.
>
> [2] J. Weston and C. Watkins. Support vector machines for multi-class pattern recognition. European Symposium on Artificial Neural Networks, 4(6), 1999.
>
> **5. The applicability of this work might be limited to learning to defer.**
>
> **Response:** While the proof techniques and methods may not extend directly to other settings beyond L2D, we believe our work connecting realizable $H$-consistency, $H$-consistency bounds, and Bayes-consistency could provide valuable insights into these consistency notions in other learning settings. In particular, these connections, along with our approach of constructively replacing indicator functions with smooth loss functions in the novel derivation of new surrogate losses from first principles, could help design loss functions that benefit from strong consistency guarantees in various scenarios.
>
> **Questions:**
>
> **1. Can constrained losses [1] be used as the base loss?**
>
> **Response:** That's an excellent question. We expect that the constrained losses can also be used to replace indicator functions in the deferral loss, based on a similar approach. Briefly, we expect that the first indicator function can be replaced by the standard constrained loss and the second indicator function can be replaced by a modified constrained loss. Establishing realizable $H$-consistency and $H$-consistency bounds for this new family of surrogate losses can be a very interesting future direction. We will elaborate on this insightful point brought up by the reviewer in the final version.
>
> **2. Lines 273-274: Can negative results be proven formally?**
>
> **Response:** We will seek to derive such counterexamples in the final version. This will involve carefully designing the conditional distribution, the expert, and the cost function to violate the more stringent condition for $H$-consistency in the additional cases due to the general cost function.

---

> > ### Comment · Reviewer_W4md · 2024-08-09
> > **Increase my rating to 7**
> >
> > Thank you for the detailed responses to my questions and concerns, as well as those in other reviews. I am satisfied with the reply and have increased my rating to 7.

---

> > > ### Author Response · Authors · 2024-08-10
> > >
> > > We are glad to have addressed the reviewer's questions and sincerely appreciate their updated rating, insightful feedback, and recognition of our work. Please let us know if there is any other question.

---

### Official Review · Reviewer_oVSD · 2024-07-11

**Soundness:** 3
**Presentation:** 3
**Contribution:** 3
**Rating:** 7
**Confidence:** 3

**Summary:**

This paper considers the problem of learning to defer (L2D),  where a classifier is allowed to defer a decision to an expert (possibly expensive to query) and trained to accurately predict while minimising the expert cost.

A major contribution of this paper is establishing consistency guarantees for surrogate losses.
Specifically, the authors derive a loss that encompasses existing surrogate losses.
The authors provide sufficient conditions for the proposed surrogate loss to have realisable H consistency and for H consistency bounds.
From the derived H consistency bounds, the authors show that certain choices of the expert cost leads to a surrogate loss that is both Bayes and realisable H consistent.

**Strengths:**

The paper is very well written and does not have major issues in clarity.

This paper seems to provide a general  framework for analysing surrogate losses for L2D, and the established results appear to be novel.
I am not actively working on L2D and my evaluation might not be accurate.

**Weaknesses:**

Due to my lack of experience in the field, I find it challenging to adequately adjudicate the significance of the work.
Realisable H consistency effectively assumes that there is a perfect classifier that does not need an expert.
This preposition seems to be strong and diminishes the point of having an expert.

**Questions:**

1. Why is H consistency important in L2D?

**Limitations:**

Yes

---

> ### Author Rebuttal · Authors · 2024-08-06
>
> Thank you for your appreciation of our work. We will take your suggestions into account when preparing the final version. Below please find responses to specific questions.
>
> **Weaknesses: Due to my lack of experience in the field, I find it challenging to adequately adjudicate the significance of the work. Realizable H consistency effectively assumes that there is a perfect classifier that does not need an expert. This preposition seems to be strong and diminishes the point of having an expert.**
>
> **Response:** The notion of realizable $H$-consistency was first studied by Long and Servedio (2013) and Zhang and Agarwal (2020) in the standard classification setting. There, it means that under the assumption there is a classifier that achieves zero multi-class zero-one loss, minimizing the surrogate loss also leads to a zero-error solution.
>
> This notion was extended to the Learning to Defer (L2D) setting by Mozannar et al. (2023). Here, it implies that under the assumption that there is a predictor (a standard classifier augmented with a deferral option) and an expert who achieve zero deferral loss, minimizing the surrogate loss also results in a zero-error solution. Note that in the L2D setting, the assumption does not imply "there is a perfect classifier that does not need an expert." Rather, it means that for every instance $x$, either deferral does not occur and the predictor achieves the zero multi-class zero-one loss, or deferral occurs but the expert achieves zero cost $c(x, y) = 0$.
>
>  When the cost is defined as the expert’s classification error: $c(x, y) = 1_{\mathsf{g}(x) \neq y}$, as in previous work by Mozannar and Sontag (2020), Verma and Nalisnick (2022), and Mozannar et al. (2023), it implies that the expert achieves a zero multi-class zero-one loss solution on the instance $x$ where deferral occurs. This does not diminish the point of having an expert. Instead, it effectively assumes that there is a good balance between a standard classifier, a deferral option, and an expert such that for every instance $x$, the deferral loss is zero.
>
> **Questions: Why is H consistency important in L2D?**
>
> **Response:** Directly optimizing the deferral loss function, which is the target loss in L2D, is generally computationally intractable for complex hypothesis sets $H$. Therefore, a common approach is to optimize a surrogate loss that facilitates the optimization of the deferral loss function. But, what guarantees can we rely on when minimizing a surrogate loss of the deferral loss? This is a fundamental question with significant implications for L2D.
>
> A desirable guarantee often referred to in this context is Bayes-consistency. It means that optimizing the surrogate loss over the family of all measurable functions leads to the minimization of the deferral loss over the same family. However, while Bayes-consistency is valuable, it is not sufficiently informative, as it is established for the family of all measurable functions and disregards the crucial role played by restricted hypothesis sets in learning, such as a family of linear models or neural networks.
>
> As pointed out by Long and Servedio [2013], in some cases, minimizing Bayes-consistent losses can result in constant expected error, while minimizing inconsistent losses can yield an expected loss approaching zero. To address this limitation, the authors introduced the concept of realizable $H$-consistency, further explored by Zhang and Agarwal [2020] and more recently, by Mozannar et al. (2023) in L2D. Nonetheless, these guarantees are only asymptotic and do not provide guarantees for approximate minimizers.
>
> Recent research by Awasthi et al. [2022b,a] has instead introduced and analyzed $H$-consistency bounds, further explored by Mao et al. [2023a, 2024a] in L2D. These bounds are more informative than Bayes-consistency since they are hypothesis set-specific and non-asymptotic. Crucially, they provide upper bounds on the estimation error of the target loss, for example, the deferral loss, that holds for any predictor $h$ within a hypothesis set $H$. These bounds relate this estimation error to the surrogate loss estimation error.
>
> Realizable $H$-consistency and $H$-consistency bounds were first proposed and extensively studied in standard classification. $H$-consistency bounds imply Bayes-consistency and realizable $H$-consistency in the standard classification setting and represent a state-of-the-art consistency guarantee since it is both non-asymptotic and hypothesis set-dependent. As with Bayes-consistency studied in a variety of scenarios, it is natural to study the stronger $H$-consistency in scenarios including L2D.
>
> Although both realizable $H$-consistency and $H$-consistency bounds have been explored in the L2D setting, no existing surrogate loss achieves realizable $H$-consistency and $H$-consistency bounds (including Bayes-consistency) simultaneously. Furthermore, the connection between these consistency notions is not as clear as that in the standard classification.
>
> Our work fills this gap by presenting a comprehensive study of these consistency notions and surrogate loss functions for L2D. We introduce a new family of surrogate losses and establish their realizable $H$-consistency and $H$-consistency bounds under different cases. Our results also resolve an open question raised in previous work [Mozannar et al., 2023] by proving the realizable H-consistency and Bayes-consistency of a specific surrogate loss. We further investigate the relationship between $H$-consistency bounds and realizable $H$-consistency in L2D, highlighting key differences from standard classification.
>
> Furthermore, these $H$-consistency guarantees provide significant *algorithmic benefits* when minimizing our new surrogate loss functions, as illustrated by our experiments. We refer the reviewer to our response to Question 1 of Reviewer Nrap for more details on the superiority of our surrogate losses compared to previous work.

---

> ### Comment · Reviewer_oVSD · 2024-08-08
>
> Thank you for your clarification. I assumed $c(x, y)$ included the cost of deferral (as in $1_{g(x)\neq y} + \beta$ with $\beta > 0$) not just the loss incurred by a mistake by the expert, which caused my confusion.
> I agree that "deferral occurs but the expert achieves zero cost" holds for a loss like $1_{g(x) \neq y}$

---

> > ### Author Response · Authors · 2024-08-09
> >
> > We are glad that the concerns have been addressed and appreciate the reviewer's support for our work. Please let us know if there is any other question.

---

### Official Review · Reviewer_kR4F · 2024-07-12

**Soundness:** 3
**Presentation:** 2
**Contribution:** 2
**Rating:** 6
**Confidence:** 3

**Summary:**

The authors provide a framework of surrogate loss functions for learning to defer under the multi-class classification problem. By examining the deferral loss function and choosing different surrogates for the indicator functions, the authors provide a novel class of surrogate loss functions for learning to defer and prove the realizable $\mathcal{H}$-consistency and $\mathcal{H}$-consistency for some cases. The authors also verify their proposed loss with numerical results.

**Strengths:**

1. The analysis is novel and provides new insights for learning to defer literature;
2. The paper addresses the problem of achieving $\mathcal{H}$-consistency and realizable $\mathcal{H}$-consistency simultaneously.

**Weaknesses:**

1. (Major) The proposed loss is not practical in many critical settings when the cost to consult an expert is high; Both terms of the derived loss function contain the exact value of $c(x, y)$ while knowing $c(x, y)$ itself needs consulting the expert if $c(x, y)$ is related to the expert's response. Thus, training a model using the proposed surrogate loss requires querying not only the true label but also the expert's response to every single sample, which is somewhat against the motivation of deriving the problem of learning to defer. See the derivation of equation (2). (Addressed, improve my rating to 6)
2. (Minor) Some of the results are restricted to the case $c(x,y) = 1_{g(x) \neq y}$, which is limited compared to the vast problem settings of learning to defer;
3. (Minor) The presentation is a little confusing: there are three consistencies (Bayes consistency, $\mathcal{H}$- consistency, and realizable $\mathcal{H}$-consistency), while their relationships should be formally summarized into some propositions; the existing surrogate loss functions' consistencies should also be summarized into one table with indicator "yes", "no", or "not proved" for each consistency.
4. (Moderate) The numerical experiments don't specify the expert algorithm; the experiments are also very restricted to the case of $c(x,y) = 1_{g(x) \neq y}$.

**Questions:**

1. The major concern of mine is that the training under the proposed loss function requires the full responses of the expert to the entire training dataset, which is undesirable for the L2D problem. Previous surrogate loss functions (nominated in Section 3.3) only require $c(x, y)$ if the model chooses to defer. The learning to defer problem is proposed to ease the pain of consulting experts, while I think the loss function proposed works against that principle. Is it possible to mitigate the training cost of the model?
2. All the remaining questions are addressed in the Weakness part and do not matter unless the first question is addressed.

---

> ### Author Rebuttal · Authors · 2024-08-06
>
> Thank you for your insightful comments. We have carefully addressed all the questions raised. Please find our responses below.
>
> **Weakness 1: The proposed loss is not practical ... the derivation of equation (2).**
>
> **Question 1. The major concern ... mitigate the training cost of the model?**
>
> **Response:** Let us begin by addressing a misconception: all previous surrogate loss functions, including those in Section 3.3, require knowing the costs $c(x, y)$ for all training points. This is essential to select an effective predictor that minimizes the need to consult experts during inference. The same requirement applies also to the original deferral loss itself (whose direct minimization is intractable).
>
> Furthermore, Eq. (2) is just an equivalent form of the original deferral loss, which helps derive new surrogate loss functions from first principles.
>
> The goal of the L2D framework is to suitably decide whether to predict a label or defer to an expert at *inference time*. During inference, it is only necessary to compute the output of the selected expert.
>
> Learning a good deferral solution requires training a predictor with an augmented label corresponding to the deferral options using the training data. Thus, the full responses of the experts are required during training to train the predictor effectively.
>
> Our work is fully consistent with the L2D framework and does not violate its principle.
>
> Now, it's certainly possible to explore alternative solutions to mitigate training costs:
>
> - Using Precomputed Costs: Instead of relying on instance-specific costs, one could use a general precomputed cost, such as the average error of an expert. This simplifies training but might lead to a less optimal deferral solution compared to using instance-dependent costs.
>
> - Learning to Predict Costs: Another approach is to train a model $f$ for each expert to predict the costs $c(x, y)$. While pre-training is also needed here, it might be possible to use a smaller training set. This approach introduces trade-offs worth investigating, depending on the desired level of cost estimation accuracy.
>
> - Partial Cost Computation: A third option involves using only a fraction of the costs during training and leveraging techniques like importance weighting to estimate the costs for instances where they weren't computed.
>
> **Weakness 2. (Minor) Some of the results are restricted to the case $c(x, y) = 1_{g(x) \neq y}$ ... settings of learning to defer.**
>
> **Response:** The case $c(x, y) = 1_{\mathsf{g}(x) \neq y}$ has been adopted in previous fundamental work on L2D [Mozannar and Sontag, 2020; Verma and Nalisnick, 2022], including the recent study on realizable $H$-consistency by Mozannar et al. (2023). The surrogate losses $\mathsf{L}_ {\mathrm{CE}}$, $\mathsf{L}_ {\mathrm{OVA}}$, and $\mathsf{L}_ {\mathrm{RS}}$ in Section 3.3 are all proposed in this context.
>
> Our work introduces a new family of surrogate losses parameterized by a function $\Phi$ that can achieve Bayes-consistency, realizable $H$-consistency, and $H$-consistency bounds simultaneously, both in the case of $c(x, y) = 1_{\mathsf{g}(x) \neq y}$ and for general cost functions with appropriate choices of $\Phi$. Our work also highlights that consistency guarantees can differ between the standard case of $c(x, y) = 1_{\mathsf{g}(x) \neq y}$ and cases involving general cost functions for surrogate loss functions in L2D, a distinction not pointed out in previous studies. Additionally, we resolve an open question raised in previous work [Mozannar et al., 2023] by proving the realizable $H$-consistency and Bayes-consistency of a specific surrogate loss in the case $c(x, y) = 1_{\mathsf{g}(x) \neq y}$.
>
> **Weakness 3. (Minor) The presentation is a little confusing ... for each consistency.**
>
> **Response:** Thank you for the suggestions. We will clarify the relationships between the three consistency notions in the propositions and include tables summarizing the properties of existing surrogate losses by using an additional page in the final version. Below is a table of the consistency properties of existing surrogate losses and ours in the case of $c(x, y) = 1_{\mathsf{g}(x) \neq y}$.
>
> | Surrogate losses| Realizable $H$-consistency| Bayes-consistency| $H$-consistency bounds|
> |-|-|-|-|
> | $\mathsf{L_{\mathrm{CE}}}$|no|yes|yes|
> | $\mathsf{L_{\mathrm{OvA}}}$|no|yes|yes|
> | $\mathsf{L_{\mathrm{general}}}$|no|yes|yes|
> | $\mathsf{L_{\mathrm{RS}}}$ ($\mathsf{L}_{\mathrm{RL2D}}$ with $\Psi(t) = -\log(t)$)|yes|yes (proved by us)|yes (proved by us)|
> | $\mathsf{L}_{\mathrm{RL2D}}$ with $\Psi(t) = \frac{1}{q} \left(1 - t^q\right), q \in (0, 1)$|yes|yes|yes|
> | $\mathsf{L}_{\mathrm{RL2D}}$ with $\Psi(t) = 1 - t$|yes|yes|yes|
>
> **Weakness 4. (Moderate) The numerical experiments don't specify the expert ... of $c(x, y) = 1_{g(x) \neq y}$.**
>
> **Response:** We follow the setting of Mozannar et al. [2023] and adopt the same expert algorithm as in [Mozannar et al., 2023, Table 1, Human column]. We will clarify this in the final version.
>
> As mentioned before, most surrogate losses proposed in previous work, except $\mathsf{L}_ {\mathrm{general}}$, are in the case of $c(x, y) = 1_{\mathsf{g}(x) \neq y}$. This naturally leads to a comparison of these surrogate losses in the context of $c(x, y) = 1_{\mathsf{g}(x) \neq y}$.
>
> We have included an additional experiment involving general cost functions. In the non-realizable case with general cost functions, the additional experimental result (Figure 2 in the global response) shows that our surrogate loss $\mathsf{L}_ {\mathrm{RL2D}}$ with $q = 1$ performs comparably to the surrogate loss $\mathsf{L}_ {\mathrm{general}}$, as both are supported by $H$-consistency bounds and Bayes-consistency with general cost functions. Our surrogate loss $\mathsf{L}_ {\mathrm{RL2D}}$ with $q = 1$ outperforms $\mathsf{L}_ {\mathrm{RS}}$ because the latter does not benefit from Bayes-consistency with general cost functions.

---

> > ### Comment · Reviewer_kR4F · 2024-08-07
> >
> > Thank the authors for their detailed reply. My major concern has been addressed, and I will improve my rating correspondingly.

---

> > > ### Author Response · Authors · 2024-08-09
> > >
> > > We are pleased to have addressed the reviewer's concerns and are grateful for their insightful comments and constructive suggestions. Please let us know if there is any other question.

---

### Official Review · Reviewer_Nrap · 2024-07-13

**Soundness:** 4
**Presentation:** 4
**Contribution:** 2
**Rating:** 6
**Confidence:** 5

**Summary:**

This paper proposes considers the setting of learning to defer: a machine learning system can choose to either classify an instance or defer the decision to an expert which incurs a variable cost. The objective is to minimize the deferral loss of the system. To solve this problem, prior work has proposed surrogate losses with certain theoretical guarantees with respect to the original deferral loss. This work proposes the first surrogate loss RL2D that is realizable H-consistent, satisfies an H-consistency bound and thus is bayes consistent. This resolves an open problem proposed from prior work. Empirically, the authors showcase that the proposed surrogate exceeds or matches prior surrogates on three different datasets.

**Strengths:**

Originality: The surrogate loss proposed in the work is novel as well as the proof technique for the theoretical properties. The derivation of the surrogate loss is different from prior work, however, it is not clear how the technique can be generalized to other settings.

Quality: I have verified a good portion of the theoretical derivations and they seem sound. The experimental setup follows similar protocol to prior work and is sound. The paper is very strong theoretically.

Clarity: very well written, clearly stating contributions of prior work and setting the stage for readers unfamiliar with the setting and the literature. Derivation and theoretical properties very well stated and easy to follow along.

Significance: The paper settles an open problem from prior work at AISTATS and in turns I believe concludes (barring any breakthroughs) a line of work on deriving surrogate losses for learning to defer. I think this is important because now the community can focus on other settings and other considerations beyond theoretical consistency properties. However, I don't think the paper has a lot to offer in terms of techniques/methods for the community beyond the learning to defer problem as the derivation rely on some algebra of the deferral loss. Therefore, I don't expect this to be widely read by the community, but will be instead read in great detail by the community working on learning to defer and related problems.

**Weaknesses:**

There are no weaknesses with regard to the theory in this paper beyond the generalizability of the approach taken to related problem settings.

However, the experimental setting is quite limited in terms of showcasing the behavior empirically of the newly proposed method. For the use of the surrogate in practice, it is not clear in which scenarios (if any) is the surrogate superior to prior work.

Moreover, it is not clear how do the theoretical properties help in practice.

**Questions:**

- In which settings is the new proposed surrogate better than prior work?

- How do the theoretical properties manifest in practice?


Follow-up after author response:
- the authors have done a good job answering the concerns, I believe this paper merits acceptance but I will maintain my original score as I am unsure about the level of impact of the paper.

**Limitations:**

Yes they have.

---

> ### Author Rebuttal · Authors · 2024-08-06
>
> Thank you for your appreciation of our work. We will take your suggestions into account when preparing the final version. Below please find responses to specific questions.
>
> **Weaknesses:**
>
> **1. There are no weaknesses with regard to the theory in this paper beyond the generalizability of the approach taken to related problem settings.**
>
> **Response:** Thank you for appreciating our theoretical contributions. While the proof techniques and methods may not extend directly to other settings beyond L2D, we believe our work connecting realizable $H$-consistency, $H$-consistency bounds, and Bayes-consistency could provide valuable insights into these consistency notions in other learning settings. In particular, these connections, along with our approach of constructively replacing indicator functions with smooth loss functions in the novel derivation of new surrogate losses from first principles, could help design loss functions that benefit from strong consistency guarantees in various scenarios.
>
> **2. However, the experimental setting is quite limited in terms of showcasing the behavior empirically of the newly proposed method.**
>
> **Response:** Thank you for the feedback. We have included two additional experiments in the global response: the realizable case and the non-realizable case with general cost functions. In the realizable case, the additional experimental result (Figure 1) shows that our surrogate loss $\mathsf L_{\mathrm{RL2D}}$ with $q = 0.7$ and $q = 1$ are realizable $H$-consistent, while $\mathsf L_{\mathrm{CE}}$, $\mathsf L_{\mathrm{OVA}}$ and $\mathsf L_{\mathrm{general}}$ are not. This validates our theory.
>
> In the non-realizable case with general cost functions, the additional experimental result (Figure 2) shows that our surrogate loss $\mathsf{L}_ {\mathrm{RL2D}}$ with $q = 1$ performs comparably to the surrogate loss $\mathsf{L}_ {\mathrm{general}}$, as both are supported by $H$-consistency bounds and Bayes-consistency with general cost functions. Our surrogate loss $\mathsf{L}_ {\mathrm{RL2D}}$ with $q = 1$ outperforms $\mathsf{L}_ {\mathrm{RS}}$ because the latter does not benefit from Bayes-consistency with general cost functions.
>
> **Questions:**
>
> **1. In which settings is the new proposed surrogate better than prior work?**
>
> **Weakness: For the use of the surrogate in practice, it is not clear in which scenarios (if any) is the surrogate superior to prior work.**
>
> **Response:** Our surrogate losses $\mathsf{L}_ {\mathrm{RL2D}}$ satisfying Theorem 4.1 perform better in realizable scenarios than the surrogate losses $\mathsf{L}_ {\mathrm{CE}}$, $\mathsf{L}_ {\mathrm{OVA}}$, and $\mathsf{L}_ {\mathrm{general}}$ from prior work, as ours are realizable $H$-consistent while theirs are not. This is illustrated by our experiment in the realizable case (Figure 1 in the global response).
>
> Our surrogate losses $\mathsf{L}_ {\mathrm{RL2D}}$ satisfying Theorem 4.2 and Corollary 4.4 are comparable to the surrogate losses in prior work in non-realizable scenarios when the cost is the expert's classification error, as all of them are Bayes-consistent and supported by $H$-consistency bounds. This is demonstrated by our experiment in the non-realizable case with the cost function being the expert's classification error (Table 2 in the submission).
>
> Our surrogate losses $\mathsf{L}_ {\mathrm{RL2D}}$ satisfying Theorem 4.3 and Corollary 4.5 are superior to the surrogate loss $\mathsf{L}_ {\mathrm{RS}}$ in non-realizable scenarios with general cost functions, as ours are supported by $H$-consistency bounds and Bayes-consistency while theirs are not. This is evidenced by our experiment in the non-realizable case with general cost functions (Figure 2 in the global response).
>
> **2. How do the theoretical properties manifest in practice?**
>
> **Weakness: Moreover, it is not clear how do the theoretical properties help in practice.**
>
> **Response:** The additional experimental result (Figure 1 in the global response) in the realizable scenario demonstrates that our realizable $H$-consistent surrogate loss $\mathsf{L}_ {\mathrm{RL2D}}$ with $q = 0.7$ and $q = 1$ outperforms $\mathsf{L}_ {\mathrm{CE}}$, $\mathsf{L}_ {\mathrm{OVA}}$, and $\mathsf{L}_ {\mathrm{general}}$, which are not realizable $H$-consistent. Realizable $H$-consistency is beneficial in practice since realizable scenarios are common, particularly in the current use of neural networks in applications. Furthermore, the simultaneous $H$-consistency properties of our surrogate losses support their use in non-realizable scenarios, as they share the same Bayes-consistency properties with existing surrogate losses and are expected to perform comparably. The additional experimental result (Figure 2 in the global response) in the non-realizable scenario with general cost functions shows that our surrogate loss $\mathsf{L}_ {\mathrm{RL2D}}$ with $q = 1$ performs comparably to the surrogate loss $\mathsf{L}_ {\mathrm{general}}$, as both are supported by H-consistency bounds and Bayes-consistency with general cost functions. Our surrogate loss $\mathsf{L}_ {\mathrm{RL2D}}$ with $q = 1$ outperforms $\mathsf{L}_ {\mathrm{RS}}$ because the latter does not benefit from Bayes-consistency with general cost functions. These experimental results align with and further validate our theory.

---

### Author Rebuttal · Authors · 2024-08-06

Dear reviewers,

We would like to express our appreciation for all your constructive suggestions and insightful comments. We have attached a PDF that includes additional experimental results for both the realizable case and the non-realizable case with general cost functions.

Figure 1 shows system accuracy versus training samples on a realizable mixture of Gaussian distributions in [Mozannar et al., 2023]. Our surrogate loss $\mathsf L_{\mathrm{RL2D}}$ with $q = 0.7$ and $q = 1$ are realizable $H$-consistent, while $\mathsf L_{\mathrm{CE}}$, $\mathsf L_{\mathrm{OVA}}$ and $\mathsf L_{\mathrm{general}}$ are not. This validates our theory.

Figure 2 shows system accuracy versus coverage on the HateSpeech dataset by varying $\beta$ in the general cost functions $c(x, y) = 1_{\mathsf g(x) \neq y} + \beta$. As $\beta$ increases, deferral algorithms yield solutions with higher coverage and decreased system accuracy. This is because $\beta$ controls the trade-off between expert's inference cost and accuracy. $\mathsf{L}_ {\mathrm{RL2D}}$ with $q = 1$ performs comparably to the surrogate loss $\mathsf{L}_ {\mathrm{general}}$, as both are supported by $H$-consistency bounds and Bayes-consistency with general cost functions. Our surrogate loss $\mathsf{L}_ {\mathrm{RL2D}}$ with $q = 1$ outperforms $\mathsf{L}_ {\mathrm{RS}}$ because the latter does not benefit from Bayes-consistency with general cost functions.

---

### Decision · Program_Chairs · 2024-09-25

**Decision:**

Accept (poster)

**Comment:**

The authors make a strong theoretical contribution for the problem of learning to defer. They propose the first surrogate loss (in fact, multiple of them) that satisfies realizable H-consistency and Bayes-consistency. This resolves an open question from Mozanner et al.’s AISTATS 2023 paper. It should have large impact in the learning to defer community as researchers in this community can move beyond consistency. All reviewers are positive on this work. It deserves to be accepted. Congratulations